# ANYTIME SAMPLING FOR AUTOREGRESSIVE MODELS VIA ORDERED AUTOENCODING

**Yilun Xu**
Massachusetts Institute of Technology
ylxu@mit.edu

**Yang Song**
Stanford University
yangsong@cs.stanford.edu

**Sahaj Garg**
Stanford University
sahajg@cs.stanford.edu

**Linyuan Gong**
UC Berkeley
gonglinyuan@hotmail.com

**Rui Shu**
Stanford University
ruishu@cs.stanford.edu

**Aditya Grover**
UC Berkeley
aditya.grover1@gmail.com

**Stefano Ermon**
Stanford University
ermon@cs.stanford.edu

## ABSTRACT

Autoregressive models are widely used for tasks such as image and audio generation. The sampling process of these models, however, does not allow interruptions and cannot adapt to real-time computational resources. This challenge impedes the deployment of powerful autoregressive models, which involve a slow sampling process that is sequential in nature and typically scales linearly with respect to the data dimension. To address this difficulty, we propose a new family of autoregressive models that enables anytime sampling. Inspired by Principal Component Analysis, we learn a structured representation space where dimensions are ordered based on their importance with respect to reconstruction. Using an autoregressive model in this latent space, we trade off sample quality for computational efficiency by truncating the generation process before decoding into the original data space. Experimentally, we demonstrate in several image and audio generation tasks that sample quality degrades gracefully as we reduce the computational budget for sampling. The approach suffers almost no loss in sample quality (measured by FID) using only 60% to 80% of all latent dimensions for image data. Code is available at https://github.com/Newbeeer/Anytime-Auto-Regressive-Model.

## 1 INTRODUCTION

Autoregressive models are a prominent approach to data generation, and have been widely used to produce high quality samples of images (Oord et al., 2016b; Salimans et al., 2017; Menick & Kalchbrenner, 2018), audio (Oord et al., 2016a), video (Kalchbrenner et al., 2017) and text (Kalchbrenner et al., 2016; Radford et al., 2019). These models represent a joint distribution as a product of (simpler) conditionals, and sampling requires iterating over all these conditional distributions in a certain order. Due to the sequential nature of this process, the computational cost will grow at least linearly with respect to the number of conditional distributions, which is typically equal to the data dimension. As a result, the sampling process of autoregressive models can be slow and does not allow interruptions.

Although caching techniques have been developed to speed up generation (Ramachandran et al., 2017; Guo et al., 2017), the high cost of sampling limits their applicability in many scenarios. For example, when running on multiple devices with different computational resources, we may wish to trade off sample quality for faster generation based on the computing power available on each device. Currently, a separate model must be trained for each device (*i.e.*, computational budget) in order to

trade off sample quality for faster generation, and there is no way to control this trade-off on the fly to accommodate instantaneous resource availability at time-of-deployment.

To address this difficulty, we consider the novel task of *adaptive* autoregressive generation under *computational constraints*. We seek to build a *single model* that can automatically trade-off sample quality versus computational cost via *anytime sampling*, *i.e.*, where the sampling process may be interrupted anytime (*e.g.*, because of exhausted computational budget) to yield a *complete* sample whose sample quality decays with the earliness of termination.

In particular, we take advantage of a generalization of Principal Components Analysis (PCA) proposed by Rippel et al. (2014), which learns an ordered representations induced by a structured application of dropout to the representations learned by an autoencoder. Such a representation encodes raw data into a latent space where dimensions are sorted based on their importance for reconstruction. Autoregressive modeling is then applied in the ordered representation space instead. This approach enables a natural trade-off between quality and computation by truncating the length of the representations: When running on devices with high computational capacity, we can afford to generate the full representation and decode it to obtain a high quality sample; when on a tighter computational budget, we can generate only the first few dimensions of the representation and decode it to a sample whose quality degrades smoothly with truncation. Because decoding is usually fast and the main computation bottleneck lies on the autoregressive part, the run-time grows proportionally relative to the number of sampled latent dimensions.

Through experiments, we show that our autoregressive models are capable of trading off sample quality and inference speed. When training autoregressive models on the latent space given by our encoder, we witness little degradation of image sample quality using only around 60% to 80% of all latent codes, as measured by Fréchet Inception Distance (Heusel et al., 2017) on CIFAR-10 and CelebA. Compared to standard autoregressive models, our approach allows the sample quality to degrade gracefully as we reduce the computational budget for sampling. We also observe that on the VCTK audio dataset (Veaux et al., 2017), our autoregressive model is able to generate the low frequency features first, then gradually refine the waveforms with higher frequency components as we increase the number of sampled latent dimensions.

## 2 BACKGROUND

**Autoregressive Models**   Autoregressive models define a probability distribution over data points $\mathbf{x} \in \mathbb{R}^D$ by factorizing the joint probability distribution as a product of univariate conditional distributions with the chain rule. Using $p_\theta$ to denote the distribution of the model, we have:

$$p_\theta(\mathbf{x}) = \prod_{i=1}^{D} p_\theta(x_i \mid x_1, \cdots, x_{i-1}) \tag{1}$$

The model is trained by maximizing the likelihood:

$$\mathcal{L} = \mathbb{E}_{p_d(\mathbf{x})}[\log p_\theta(\mathbf{x})], \tag{2}$$

where $p_d(\mathbf{x})$ represents the data distribution.

Different autoregressive models adopt different orderings of input dimensions and parameterize the conditional probability $p_\theta(x_i \mid x_1, \cdots, x_{i-1}), i = 1, \cdots, D$ in different ways. Most architectures over images order the variables $x_1, \cdots, x_D$ of image $\mathbf{x}$ in *raster scan order* (*i.e.*, left-to-right then top-to-bottom). Popular autoregressive architectures include MADE (Germain et al., 2015), PixelCNN (Oord et al., 2016b; van den Oord et al., 2016; Salimans et al., 2017) and Transformer (Vaswani et al., 2017), where they respectively use masked linear layers, convolutional layers and self-attention blocks to ensure that the output corresponding to $p_\theta(x_i \mid x_1, \cdots, x_{i-1})$ is oblivious of $x_i, x_{i+1}, \cdots, x_D$.

**Cost of Sampling**   During training, we can evaluate autoregressive models efficiently because $x_1, \cdots, x_D$ are provided by data and all conditionals $p(x_i \mid x_1, \cdots, x_{i-1})$ can be computed in parallel. In contrast, sampling from autoregressive models is an inherently sequential process and cannot be easily accelerated by parallel computing: we first need to sample $x_1$, after which we sample $x_2$ from $p_\theta(x_2 \mid x_1)$ and so on—the $i$-th variable $x_i$ can only be obtained after we have already computed $x_1, \cdots, x_{i-1}$. Thus, the run-time of autoregressive generation grows *at least linearly* with respect to the length of a sample. In practice, the sample length $D$ can be more than hundreds of

thousands for real-world image and audio data. This poses a major challenge to fast autoregressive generation on a small computing budget.

# 3 ANYTIME SAMPLING WITH ORDERED AUTOENCODERS

Our goal is to circumvent the non-interruption and linear time complexity of autoregressive models by pushing the task of autoregressive modeling from the original data space (e.g., pixel space) into an ordered representation space. In doing so, we develop a new class of autoregressive models where premature truncation of the autoregressive sampling process leads to the generation of a *lower quality* sample instead of an *incomplete* sample. In this section, we shall first describe the learning of the ordered representation space via the use of an *ordered autoencoder*. We then describe how to achieve anytime sampling with ordered autoencoders.

## 3.1 ORDERED AUTOENCODERS

Consider an autoencoder that encodes an input $\mathbf{x} \in \mathbb{R}^D$ to a code $\mathbf{z} \in \mathbb{R}^K$. Let $\mathbf{z} = e_\theta(\mathbf{x}) : \mathbb{R}^D \to \mathbb{R}^K$ be the encoder parameterized by $\theta$ and $\mathbf{x}' = d_\phi(\mathbf{z}) : \mathbb{R}^K \to \mathbb{R}^D$ be the decoder parameterized by $\phi$. We define $e_\theta(\cdot)_{\leq i} : \mathbf{x} \in \mathbb{R}^D \mapsto (z_1, z_2, \cdots, z_i, 0, \cdots, 0)^{\mathsf{T}} \in \mathbb{R}^K$, which truncates the representation to the first $i$ dimensions of the encoding $\mathbf{z} = e_\theta(\mathbf{x})$, masking out the remainder of the dimensions with a zero value. We define the ordered autoencoder objective as

$$\frac{1}{N} \sum_{i=1}^{N} \frac{1}{K} \sum_{j=1}^{K} \|\mathbf{x}_i - d_\phi(e_\theta(\mathbf{x}_i)_{\leq j})\|_2^2. \tag{3}$$

We note that Eq. (3) is equivalent to Rippel et al. (2014)'s nested dropout formulation using a uniform sampling of possible truncations. Moreover, when the encoder/decoder pair is constrained to be a pair of orthogonal matrices up to a transpose, then the optimal solution in Eq. (3) recovers PCA.

### 3.1.1 THEORETICAL ANALYSIS

Rippel et al. (2014)'s analysis of the ordered autoencoder is limited to linear/sigmoid encoder and a linear decoder. In this section, we extend the analysis to general autoencoder architectures by employing an information-theoretic framework to analyze the importance of the $i$-th latent code to reconstruction for ordered autoencoders. We first reframe our problem from a probabilistic perspective. In lieu of using deterministic autoencoders, we assume that both the encoder and decoder are stochastic functions. In particular, we let $q_{e_\theta}(\mathbf{z} \mid \mathbf{x})$ be a probability distribution over $\mathbf{z} \in \mathbb{R}^K$ conditioned on input $\mathbf{x}$, and similarly let $p_{d_\phi}(\mathbf{x} \mid \mathbf{z})$ be the stochastic counterpart to $d_\phi(\mathbf{z})$. We then use $q_{e_\theta}(\mathbf{z} \mid \mathbf{x})_{\leq i}$ to denote the distribution of $(z_1, z_2, \cdots, z_i, 0, \cdots, 0)^{\mathsf{T}} \in \mathbb{R}^K$, where $\mathbf{z} \sim q_{e_\theta}(\mathbf{z} \mid \mathbf{x})$, and let $p_{d_\phi}(\mathbf{x} \mid \mathbf{z})_{\leq i}$ represent the distribution of $p_{d_\phi}(\mathbf{x} \mid (z_1, z_2, \cdots, z_i, 0, \cdots, 0)^{\mathsf{T}} \in \mathbb{R}^K)$. We can modify Eq. (3) to have the following form:

$$\mathbb{E}_{x \sim p_d(\mathbf{x}), i \sim \mathcal{U}\{1, K\}} \mathbb{E}_{\mathbf{z} \sim q_{e_\theta}(\mathbf{z}|\mathbf{x})_{\leq i}}[-\log p_{d_\phi}(\mathbf{x}|\mathbf{z})_{\leq i}], \tag{4}$$

where $\mathcal{U}\{1, K\}$ denotes a uniform distribution over $\{1, 2, \cdots, K\}$, and $p_d(\mathbf{x})$ represents the data distribution. We can choose both the encoder and decoder to be fully factorized Gaussian distributions with a fixed variance $\sigma^2$, then Eq. (13) can be simplified to

$$\mathbb{E}_{p_d(\mathbf{x})} \left[ \frac{1}{K} \sum_{i=1}^{K} \mathbb{E}_{\mathbf{z} \sim \mathcal{N}(e_\theta(\mathbf{x})_{\leq i}; \sigma^2)} \left[ \frac{1}{2\sigma^2} \|\mathbf{x} - d_\phi(\mathbf{z})_{\leq i}\|_2^2 \right] \right].$$

The stochastic encoder and decoder in this case will become deterministic when $\sigma \to 0$, and the above equation will yield the same encoder/decoder pair as Eq. (3) when $\sigma \to 0$ and $N \to \infty$.

The optimal encoders and decoders that minimize Eq. (13) satisfy the following property.

**Theorem 1.** *Let $\mathbf{x}$ denote the input random variable. Assuming both the encoder and decoder are optimal in terms of minimizing Eq. (13), and $\forall i \in 3, \cdots, K, z_{i-1} \perp z_i \mid \mathbf{x}, z_{\leq i-2}$, we have*

$$\forall i \in \{3, \cdots, K\} : \quad I(z_i; \mathbf{x}|z_{\leq i-1}) \leq I(z_{i-1}; \mathbf{x}|z_{\leq i-2}),$$

*where $z_{\leq i}$ denotes $(z_1, z_2, \cdots, z_i)$.*

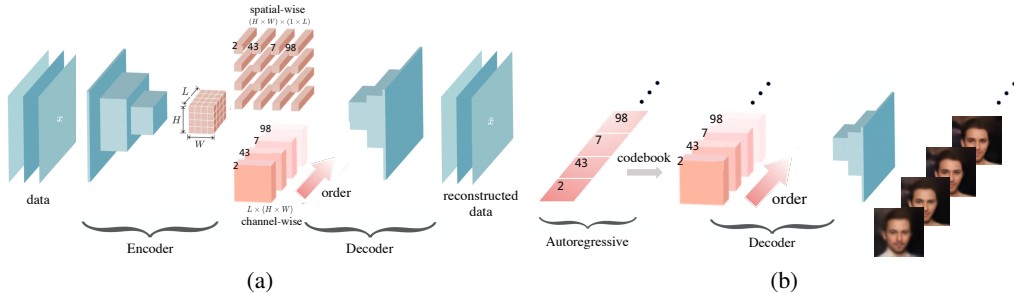

Figure 1: **(a)** Spatial-wise quantization vs. channel-wise quantization. **(b)** Anytime sampling for OVQ-VAE.

We defer the proof to Appendix A.1. The assumption $z_{i-1} \perp z_i \mid \mathbf{x}, z_{\leq i-2}$ holds whenever the encoder $q_\theta(\mathbf{z} \mid \mathbf{x})$ is a factorized distribution, which is a common choice in variational autoencoders (Kingma & Welling, 2013), and we use $I(\mathbf{a}; \mathbf{b} \mid \mathbf{c})$ to denote the mutual information between random variables $\mathbf{a}$ and $\mathbf{b}$ conditioned on $\mathbf{c}$. Intuitively, the above theorem states that for optimal encoders and decoders that minimize Eq. (13), one can extract less additional information about the raw input as the code gets longer. Therefore, there exists a natural ordering among different dimensions of the code based on the additional information they can provide for reconstructing the inputs.

## 3.2 ANYTIME SAMPLING

Once we have learned an ordered autoencoder, we then train an autoregressive model on the full length codes in the ordered representation space, also referred to as *ex-post density estimation* (Ghosh et al., 2020). For each input $\mathbf{x}_i$ in a dataset $\mathbf{x}_1, \cdots, \mathbf{x}_N$, we feed it to the encoder to get $\mathbf{z}_i = e_\theta(\mathbf{x}_i)$. The resulting codes $\mathbf{z}_1, \mathbf{z}_2, \cdots, \mathbf{z}_N$ are used as training data. After training both the ordered autoencoder and autoregressive model, we can perform anytime sampling on a large spectrum of computing budgets. Suppose for example we can afford to generate $T$ code dimensions from the autoregressive model, denoted as $\mathbf{z}_{\leq T} \in \mathbb{R}^T$. We can simply zero-pad it to get $(z_1, z_2, \cdots, z_T, 0, \cdots, 0)^\mathsf{T} \in \mathbb{R}^K$ and decode it to get a complete sample. Unlike the autoregressive part, the decoder has access to all dimensions of the latent code at the same time and can decode in parallel. The framework is shown in Fig. 1(b). For the implementation, we use the ordered VQ-VAE (Section 4) as the ordered autoencoder and the Transformer (Vaswani et al., 2017) as the autoregressive model. On modern GPUs, the code length has minimal effect on the run-time of decoding, as long as the decoder is not itself autoregressive (see empirical verifications in Section 5.2.2).

## 4 ORDERED VQ-VAE

In this section, we apply the ordered autoencoder framework to the vector quantized variational autoencoder (VQ-VAE) and its extension (van den Oord et al., 2017; Razavi et al., 2019). Since these models are quantized autoencoders paired with a latent autoregressive model, they admit a natural extension to *ordered* VQ-VAEs (OVQ-VAEs) under our framework—a new family of VQ-VAE models capable of anytime sampling. Below, we begin by describing the VQ-VAE, and then highlight two key design choices (ordered discrete codes and channel-wise quantization) critical for OVQ-VAEs. We show that, with small changes of the original VQ-VAE, these two choices can be applied straightforwardly.

### 4.1 VQ-VAE

To construct a VQ-VAE with code length of $K$ discrete latent variables , the encoder must first map the raw input $\mathbf{x}$ to a continuous representation $\mathbf{z}^e = e_\theta(\mathbf{x}) \in \mathbb{R}^{K \times D}$, before feeding it to a vector-valued quantization function $q : \mathbb{R}^{K \times D} \to \{1, 2, \cdots, C\}^K$ defined as

$$q(\mathbf{z}^e)_j = \underset{i \in \{1, \cdots, C\}}{\arg\min} \left\| \mathbf{e}_i - \mathbf{z}_j^e \right\|_2,$$

where $q(\mathbf{z}^e)_j \in \{1, 2, \cdots, C\}$ denotes the $j$-th component of the vector-valued function $q(\mathbf{z}^e)$, $\mathbf{z}_j^e \in \mathbb{R}^D$ denotes the $j$-th row of $\mathbf{z}^e$, and $\mathbf{e}_i$ denotes the $i$-th row of the embedding matrix $\mathbf{E} \in \mathbb{E}^{C \times D}$. Next, we view $q(\mathbf{z}^e)$ as a sequence of indices and use them to look up embedding vectors from the codebook $\mathbf{E}$. This yields a latent representation $\mathbf{z}_d \in \mathbb{R}^{K \times D}$, given by $\mathbf{z}_j^d = \mathbf{e}_{q(\mathbf{z}^e)_j}$, where $\mathbf{z}_j^d \in \mathbb{R}^D$ denotes the $j$-th row of $\mathbf{z}^d$. Finally, we can decode $\mathbf{z}^d$ to obtain the reconstruction $d_\phi(\mathbf{z}^d)$. This procedure can be viewed as a regular autoencoder with a non-differentiable nonlinear function that maps each latent vector $\mathbf{z}_j^e$ to 1-of-$K$ embedding vectors $\mathbf{e}_i$.

During training, we use the straight-through gradient estimator (Bengio et al., 2013) to propagate gradients through the quantization function, *i.e.*, gradients are directly copied from the decoder input $\mathbf{z}^d$ to the encoder output $\mathbf{z}^e$. The loss function for training on a single data point $\mathbf{x}$ is given by

$$\left\| d_\phi(\mathbf{z}^d) - \mathbf{x} \right\|_2^2 + \left\| \mathbf{sg}[e_\theta(\mathbf{x})] - \mathbf{z}^d \right\|_F^2 + \beta \left\| e_\theta(\mathbf{x}) - \mathbf{sg}[\mathbf{z}^d] \right\|_F^2, \tag{5}$$

where $\mathbf{sg}$ stands for the `stop_gradient` operator, which is defined as identity function at forward computation and has zero partial derivatives at backward propagation. $\beta$ is a hyper-parameter ranging from 0.1 to 2.0. The first term of Eq. (5) is the standard reconstruction loss, the second term is for embedding learning while the third term is for training stability (van den Oord et al., 2017). Samples from a VQ-VAE can be produced by first training an autoregressive model on its latent space, followed by decoding samples from the autoregressive model into the raw data space.

## 4.2 ORDERED DISCRETE CODES

Since the VQ-VAE outputs a sequence of discrete latent codes $q(\mathbf{z}^e)$, we wish to impose an ordering that prioritizes the code dimensions based on importance to reconstruction. In analogy to Eq. (3), we can modify the reconstruction error term in Eq. (5) to learn ordered latent representations. The modified loss function is an order-inducing objective given by

$$\frac{1}{K} \sum_{i=1}^{K} \left[ \left\| d_\phi(\mathbf{z}_{\leq i}^d) - \mathbf{x} \right\|_2^2 + \left\| \mathbf{sg}[e_\theta(\mathbf{x})_{\leq i}] - \mathbf{z}_{\leq i}^d \right\|_F^2 + \beta \left\| e_\theta(\mathbf{x})_{\leq i} - \mathbf{sg}[\mathbf{z}_{\leq i}^d] \right\|_F^2 \right], \tag{6}$$

where $e_\theta(\mathbf{x})_{\leq i}$ and $\mathbf{z}_{\leq i}^d$ denote the results of keeping the top $i$ rows of $e_\theta(\mathbf{x})$ and $\mathbf{z}^d$ and then masking out the remainder rows with zero vectors. We uniformly sample the masking index $i \sim \mathcal{U}\{1, K\}$ to approximate the average in Eq. (6) when $K$ is large. An alternative sampling distribution is the geometric distribution (Rippel et al., 2014). In Appendix C.3, we show that OVQ-VAE is sensitive to the choice of the parameter in geometric distribution. Because of the difficulty, Rippel et al. (2014) applies additional tricks. This results in the learning of ordered discrete latent variables, which can then be paired with a latent autoregressive model for anytime sampling.

## 4.3 CHANNEL-WISE QUANTIZATION

In Section 4.1, we assume the encoder output to be a $K \times D$ matrix (*i.e.*, $\mathbf{z}^e \in \mathbb{R}^{K \times D}$). In practice, the output can have various sizes depending on the encoder network, and we need to reshape it to a two-dimensional matrix. For example, when encoding images, the encoder is typically a 2D convolutional neural network (CNN) whose output is a 3D latent feature map of size $L \times H \times W$. Here $L$, $H$, and $W$ stand for the channel, height, and width of the feature maps. We discuss below how the reshaping procedure can significantly impact the performance of anytime sampling and propose a reshaping procedure that facilitates high-performance anytime sampling.

Consider convolutional encoders on image data, where the output feature map has a size of $L \times H \times W$. The most common way of reshaping this 3D feature map, as in van den Oord et al. (2017), is to let $H \times W$ be the code length, and let the number of channels $L$ be the size of embedding vectors, *i.e.*, $K = H \times W$ and $D = L$. We call this pattern *spatial-wise quantization*, as each spatial location in the feature map corresponds to one code dimension and will be quantized separately. Since the code dimensions

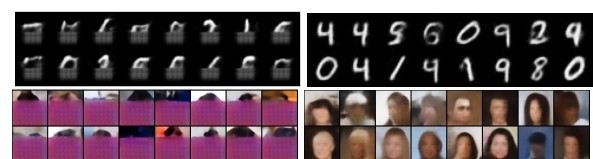

Figure 2: MNIST (**top**) and CelebA (**bottom**) samples generated with $1/4$ of the code length. **Left:** Spatial-wise quantization. **Right:** Channel-wise quantization.

correspond to spatial locations of the feature map, they encode local features due to a limited receptive field. This is detrimental to anytime sampling, because early dimensions cannot capture the global information needed for reconstructing the entire image. We demonstrate this in Fig. 2, which shows that OVQ-VAE with spatial-wise quantization is only able to reconstruct the top rows of an image with $1/4$ of the code length.

To address this issue, we propose *channel-wise quantization*, where each channel of the feature map is viewed as one code dimension and quantized separately (see Fig. 1(a) for visual comparison of spatial-wise and channel-wise quantization). Specifically, the code length is $L$ (*i.e.*, $K = L$), and the size of the embedding vectors is $H \times W$ (*i.e.*, $D = H \times W$). In this case, one code dimension includes all spatial locations in the feature map and can capture global information better. As shown in the right panel of Fig. 2, channel-wise quantization clearly outperforms spatial-wise quantization for anytime sampling. We use channel-wise quantization in all subsequent experiments. Note that in practice we can easily apply the channel-wise quantization on VQ-VAE by changing the code length from $H \times W$ to $L$, as shown in Fig. 1(a).

## 5 EXPERIMENTS

In our experiments, we focus on anytime sampling for autoregressive models trained on the latent space of OVQ-VAEs, as shown in Fig. 1(b). We first verify that our learning objectives in Eq. (3), Eq. (6) are effective at inducing ordered representations. Next, we demonstrate that our OVQ-VAE models achieve comparable sample quality to regular VQ-VAEs on several image and audio datasets, while additionally allowing a graceful trade-off between sample quality and computation time via anytime sampling. Due to limits on space, we defer the results of audio generation to Appendix B and provide additional experimental details, results and code links in Appendix E and G.

### 5.1 ORDERED VERSUS UNORDERED CODES

Our proposed ordered autoencoder framework learns an ordered encoding that is in contrast to the encoding learned by a standard autoencoder (which we shall refer to as unordered). In addition to the theoretical analysis in Section 3.1.1, we provide further empirical analysis to characterize the difference between ordered and unordered codes. In particular, we compare the importance of the $i$-th code—as measured by the reduction in reconstruction error $\Delta(i)$—for PCA, standard (unordered) VQ-VAE, and ordered VQ-VAE. For VQ-VAE and ordered VQ-VAE, we define the reduction in reconstruction error $\Delta(i)$ for a data point $\mathbf{x}$ as

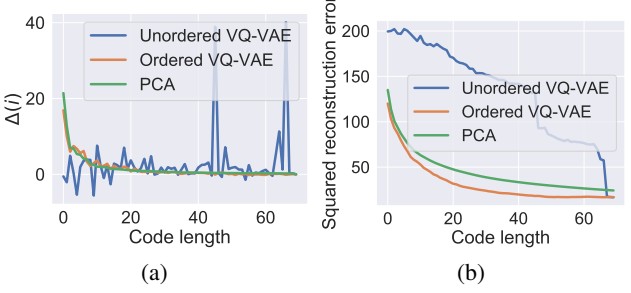

Figure 3: Ordered vs. unordered codes on CIFAR-10.

$$\Delta_{\mathbf{x}}(i) \triangleq \left\| d_\phi(\mathbf{z}_{\leq i-1}^d) - \mathbf{x} \right\|_F^2 - \left\| d_\phi(\mathbf{z}_{\leq i}^d) - \mathbf{x} \right\|_F^2 \tag{7}$$

Averaging $\Delta_{\mathbf{x}}(i)$ over the entire dataset thus yields $\Delta(i)$. Similarly we define $\Delta(i)$ as the reduction on reconstruction error of the entire dataset, when adding the $i$-th principal component for PCA.

Fig. 3(a) shows the $\Delta(i)$'s of the three models on the CIFAR-10. Since PCA and ordered VQ-VAE both learn an ordered encoding, their $\Delta(i)$'s decay gradually as $i$ increases. In contrast, the standard VQ-VAE with an unordered encoding exhibits a highly irregular $\Delta(i)$, indicating no meaningful ordering of the dimensions.

Fig. 3(b) further shows how the reconstruction error decreases as a function of the truncated code length for the three models. Although unordered VQ-VAE and ordered VQ-VAE achieve similar reconstruction errors for sufficiently large code lengths, it is evident that an ordered encoding achieves significantly better reconstructions when the code length is aggressively truncated. When sufficiently truncated, we observe even PCA outperforms unordered VQ-VAE despite the latter being a more expressive model. In contrast, ordered VQ-VAE achieves superior reconstructions compared to PCA and unordered VQ-VAE across all truncation lengths.

We repeat the experiment on standard VAE to disentangle the specific implementations in VQ-VAE. We observe that the order-inducing objective has consistent results on standard VAE (Appendix C.1).

## 5.2 Image Generation

We test the performance of anytime sampling using OVQ-VAEs (**Anytime + ordered**) on several image datasets. We compare our approach to two baselines. One is the original VQ-VAE model proposed by van den Oord et al. (2017) without anytime sampling. The other is using anytime sampling with unordered VQ-VAEs (**Anytime + unordered**), where the models have the same architectures as ours but are trained by minimizing Eq. (5). We empirically verify that 1) we are able to generate high quality image samples; 2) image quality degrades gracefully as we reduce the sampled code length for anytime sampling; and 3) anytime sampling improves the inference speed compared to naïve sampling of original VQ-VAEs.

We evaluate the model performance on the MNIST, CIFAR-10 (Krizhevsky, 2009) and CelebA (Liu et al., 2014) datasets. For CelebA, the images are resized to $64 \times 64$. All pixel values are scaled to the range $[0, 1]$. We borrow the model architectures and optimizers from van den Oord et al. (2017). The full code length and the codebook size are 16 and 126 for MNIST, 70 and 1000 for CIFAR-10, and 100 and 500 for CelebA respectively. We train a Transformer (Vaswani et al., 2017) on our VQ-VAEs, as opposed to the PixelCNN model used in van den Oord et al. (2017). PixelCNNs use standard 2D convolutional layers to capture a bounded receptive field and model the conditional dependence. Transformers apply attention mechanism and feed forward network to model the conditional dependence of 1D sequence. Transformers are arguably more suitable for channel-wise quantization, since there are no 2D spatial relations among different code dimensions that can be leveraged by convolutional models (such as PixelCNNs). Our experiments on different autoregressive models in Appendix C.2 further support the arguments.

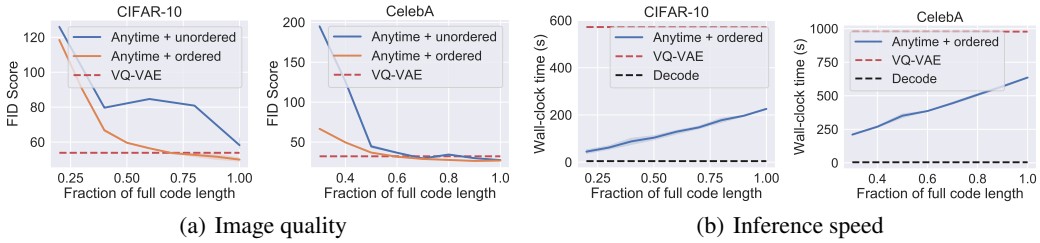

(a) Image quality          (b) Inference speed

Figure 4: (a) FID scores for anytime sampling using various code lengths. (b) Inference speed of anytime sampling with different code lengths.

### 5.2.1 Image Quality

In Fig. 4(a), we report FID (Heusel et al., 2017) scores (lower is better) on CIFAR-10 and CelebA when performing anytime sampling for ordered versus unordered VQ-VAE. FID (Fréchet Inception Distance) score is the Fréchet distance between two multivariate Gaussians, whose means and covariances are estimated from the 2048-dimensional activations of the Inception-v3 (Szegedy et al., 2016) network for real and generated samples respectively. As a reference, we also report the FID scores when using the original VQ-VAE model (with residual blocks and spatial-wise quantization) sampled at the full code length (van den Oord et al., 2017). Our main finding is that OVQ-VAE achieves a better FID score than unordered VQ-VAE at all fractional code lengths (ranging from 20% to 100%); in other words, OVQ-VAE achieves strictly superior anytime sampling performance compared to unordered VQ-VAE on both CIFAR-10 and CelebA. On CIFAR-10 dataset, a better FID score is achieved by OVQ-VAE even when sampling full codes. In Appendix C.4, we show that the regularization effect of the ordered codes causes this phenomenon.

In Fig. 5 (more in Appendix G.1), we visualize the sample quality degradation as a function of fractional code length when sampling from the OVQ-VAE. We observe a consistent increase in sample quality as we increased the fractional code length. In particular, we observe the model to initially generate a global structure of an image and then gradually fill in local details. We further show in Appendix F that, samples sharing the highest priority latent code have similar global structure.

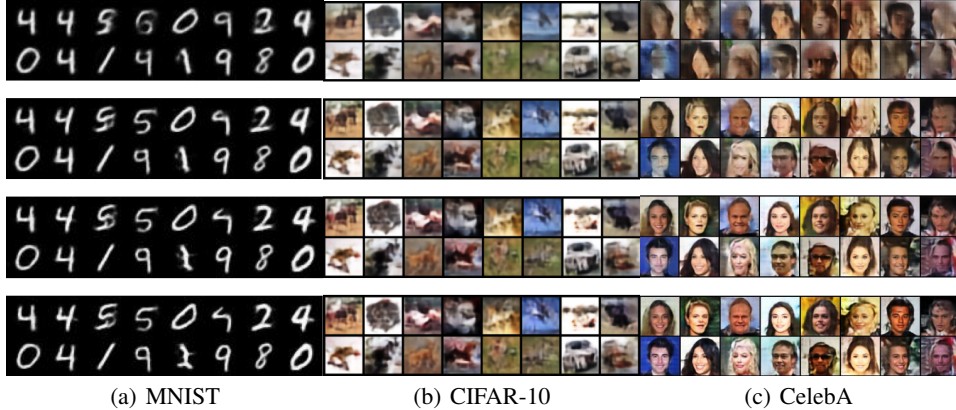

|     (a) MNIST     |     (b) CIFAR-10     |     (c) CelebA     |

Figure 5: Anytime sampling with 0.25/0.5/0.75/1.0 (**top to bottom**) fractions of full code length on MNIST (**left**), CIFAR-10 (**middle**) and CelebA (**right**) datasets.

Although our method was inspired by PCA, we encountered limited success when training an autoregressive model on the PCA-represented data. Please refer to Appendix D for more details.

### 5.2.2 INFERENCE SPEED

We compare the inference speed of our approach vs. the original VQ-VAE model by the wall-clock time needed for sampling. We also include the decoding time in our approach. We respectively measure the time of generating 50000 and 100000 images on CIFAR-10 and CelebA datasets, with a batch size of 100. All samples are produced on a single NVIDIA TITAN Xp GPU.

Fig. 4(b) shows that the time needed for anytime sampling increases almost linearly with respect to the sampled code length. This supports our argument in Section 3.2 that the decoding time is negligible compared to the autoregressive component. Indeed, the decoder took around 24 seconds to generate all samples for CelebA, whereas the sampling time of the autoregressive model was around 610 seconds—over an order of magnitude larger. Moreover, since we can achieve roughly the highest sample quality with only 60% of the full code length on CelebA, anytime sampling can save around 40% run-time compared to naïve sampling without hurting sample quality.

In addition, our method is faster than the original VQ-VAE even when sampling the full code length, without compromising sample quality (*cf*., Section 5.2.1). This is because the Transformer model we used is sufficiently shallower than the Gated PixelCNN (van den Oord et al., 2016) model in the original VQ-VAE paper. Compared to PixelCNN++ (Salimans et al., 2017), an autoregressive model on the raw pixel space, the sampling speed of our method can be an order of magnitude faster since our autoregressive models are trained on the latent space with much lower dimensionality.

## 6 RELATED WORK

Prior work has tackled the issue of slow autoregressive generation by improving implementations of the generation algorithm. For example, the sampling speed of convolutional autoregressive models can be improved substantially by caching hidden state computation (Ramachandran et al., 2017). While such approaches provide substantial speedups in generation time, they are still at best linear in the dimension of the sample space. van den Oord et al. (2018) improves the inference speed by allowing parallel computing. Compared to our approach, they do not have the test-time adaptivity to computational constraints. In contrast, we design methods that allow trade-offs between generation speed and sample quality on-the-fly based on computational constraints. For example, running apps can accommodate to the real-time computational resources without model re-training. In addition, they can be combined together with our method without sacrificing sample quality. Specifically, Ramachandran et al. (2017) leverages caches to speed up autoregressive sampling, which can be directly applied to our autoregressive model on ordered codes without affecting sample quality. van den Oord et al. (2018) proposes probability density distillation to distill autoregressive models

into fast implicit generators. We can apply the same technique on our latent autoregressive model to allow a similar speedup.

In order to enable anytime sampling, our method requires learning an ordered latent representation of data by training ordered autoencoders. Rippel et al. (2014) proposes a generalization of Principal Components Analysis to learn an ordered representation. Instead of the uniform distribution over discrete codes in our method, they adopted a geometric distribution over continuous codes during the training of the ordered autoencoders. Because of the difference they require additional tricks such as unit sweeping and adaptive regularization coefficients to stabilize the training, while our method is more stable and scalable. In addition, they only focus on fast retrieval and image compression. By contrast, we further extend our approach to autoencoders with discrete latent codes (*e.g.*, VQ-VAEs) and explore their applications in anytime sampling for autoregressive models. Another work related to our approach is hierarchical nonlinear PCA (Scholz & Vigário, 2002). We generalize their approach to latent spaces of arbitrary dimensionality, and leverage Monte Carlo estimations to improve the efficiency when learning very high dimensional latent representations. The denoising generative models proposed by Song & Ermon (2019); Ho et al. (2020) progressively denoise images into better quality, instead of modeling images from coarse to fine like our methods. This means that interrupting the sampling procedure of diffusion models at an early time might lead to very noisy samples, but in our case it will lead to images with corrector coarse structures and no noise which is arguably more desirable.

A line of works draw connections between ordered latent codes and the linear autoencoders. Kunin et al. (2019) proves that the principal directions can be deduced from the critical points of $L_2$ regularized linear autoencoders, and Bao et al. (2020) further shows that linear autoencoders can directly learn the ordered, axis-aligned principal components with non-uniform $L_2$ regularization.

## 7 CONCLUSION

Sampling from autoregressive models is an expensive sequential process that can be intractable when on a tight computing budget. To address this difficulty, we consider the novel task of adaptive autoregressive sampling that can naturally trade-off computation with sample quality. Inspired by PCA, we adopt ordered autoencoders, whose latent codes are prioritized based on their importance to reconstruction. We show that it is possible to do anytime sampling for autoregressive models trained on these ordered latent codes—we may stop the sequential sampling process at any step and still obtain a complete sample of reasonable quality by decoding the partial codes.

With both theoretical arguments and empirical evidence, we show that ordered autoencoders can induce a valid ordering that facilitates anytime sampling. Experimentally, we test our approach on several image and audio datasets by pairing an ordered VQ-VAE (a powerful autoencoder architecture) and a Transformer (an expressive autoregressive model) on the latent space. We demonstrate that our samples suffer almost no loss of quality (as measured by FID scores) for images when using only 60% to 80% of all code dimensions, and the sample quality degrades gracefully as we gradually reduce the code length.

### ACKNOWLEDGEMENTS

We are grateful to Shengjia Zhao, Tommi Jaakkola and anonymous reviewers in ICLR for helpful discussion. We would like to thank Tianying Wen for the lovely figure. YX was supported by the HDTV Grand Alliance Fellowship. SE was supported by NSF (#1651565, #1522054, #1733686), ONR (N00014- 19-1-2145), AFOSR (FA9550-19-1-0024), Amazon AWS, and FLI. YS was partially supported by the Apple PhD Fellowship in AI/ML. This work was done in part while AG was a Research Fellow at the Simons Institute for the Theory of Computing.

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

# A PROOFS

## A.1 PROOF FOR THEOREM 1

*Proof.* For simplicity we denote the distribution of the stochastic part (first $i$ dimensions) of $q_{e_\theta}(\mathbf{z}|\mathbf{x})_{\leq i}$ as $q_{e_\theta}(z_{\leq i}|\mathbf{x})$, and similarly we denote $p_{d_\phi}(\mathbf{x}|\mathbf{z})_{\leq i}$ as $p_{d_\phi}(\mathbf{x}|z_{\leq i})$. We first reformulate the objective

$$
\begin{aligned}
\mathcal{L} &= -\mathbb{E}_{x \sim p_d(\mathbf{x}), i \sim \mathcal{U}\{1,K\}} \mathbb{E}_{\mathbf{z} \sim q_{e_\theta}(\mathbf{z}|\mathbf{x})_{\leq i}} [\log p_{d_\phi}(\mathbf{x}|\mathbf{z})_{\leq i}] \\
&= \frac{-1}{K} \sum_{i=1}^{K} \mathbb{E}_{\mathbf{x} \sim p_d(\mathbf{x})} \mathbb{E}_{\mathbf{z} \sim q_{e_\theta}(\mathbf{z}|\mathbf{x})_{\leq i}} [\log p_{d_\phi}(\mathbf{x}|\mathbf{z})_{\leq i}] \\
&= \frac{-1}{K} \sum_{i=1}^{K} \int p_d(\mathbf{x}) q_{e_\theta}(z_{\leq i}|\mathbf{x}) \log p_{d_\phi}(\mathbf{x}|z_{\leq i}) d\mathbf{x} dz_{\leq i} \\
&\geq \frac{-1}{K} \sum_{i=1}^{K} \int p_d(\mathbf{x}) q_{e_\theta}(z_{\leq i}|\mathbf{x}) \log q_{e_\theta}(\mathbf{x}|z_{\leq i}) d\mathbf{x} dz_{\leq i}
\end{aligned}
\tag{8}
$$

The inequality (8) holds because KL-divergences are non-negative. Under the assumption of an optimal decoder $\phi$, we can achieve the equality in (8), in which case the objective equals

$$
\mathcal{L} = \frac{-1}{K} \sum_{i=1}^{K} \int q_{e_\theta}(z_{\leq i}, \mathbf{x}) \log \frac{q_{e_\theta}(\mathbf{x}, z_{\leq i})}{q_{e_\theta}(z_{\leq i})} d\mathbf{x} dz_{\leq i}
$$

We can define the following modified objective by adding the data entropy term, since it is a constant independent of $\theta$.

$$
\begin{aligned}
\mathcal{L} &= \frac{-1}{K} \sum_{i=1}^{K} \int q_{e_\theta}(z_{\leq i}, \mathbf{x}) \log \frac{q_{e_\theta}(\mathbf{x}, z_{\leq i})}{q_{e_\theta}(z_{\leq i}) p_d(\mathbf{x})} d\mathbf{x} dz_{\leq i} \\
&= \frac{-1}{K} \sum_{i=1}^{K} I(\mathbf{x}; z_{\leq i})
\end{aligned}
\tag{9}
$$

If there exists an integer $i \in \{2, \cdots, K\}$ such that $I(z_i; \mathbf{x}|z_{\leq i-1}) > I(z_{i-1}; \mathbf{x}|z_{\leq i-2})$, we can exchange the position of $z_i$ and $z_{i-1}$ to increase the value of objective (9). By the chain rule of mutual information we have:

$$
\begin{aligned}
I(\mathbf{x}; z_i, z_{i-1}, z_{\leq i-2}) &= I(z_{i-1}, z_{\leq i-2}; \mathbf{x}) + I(z_i; \mathbf{x}|z_{\leq i-1}) \\
&= I(z_i, z_{\leq i-2}; \mathbf{x}) + I(z_{i-1}; \mathbf{x}|z_{\leq i-2}, z_i).
\end{aligned}
$$

When $I(z_i; \mathbf{x}|z_{\leq i-1}) > I(z_{i-1}; \mathbf{x}|z_{\leq i-2})$, we can show that $I(\mathbf{x}; z_i, z_{\leq i-2}) > I(\mathbf{x}; z_{i-1}, z_{\leq i-2})$:

$$
\begin{aligned}
&I(\mathbf{x}; z_i, z_{\leq i-2}) - I(\mathbf{x}; z_{i-1}, z_{\leq i-2}) \\
&= I(z_i; \mathbf{x}|z_{\leq i-1}) - I(z_{i-1}; \mathbf{x}|z_{\leq i-2}, z_i) \\
&> I(z_{i-1}; \mathbf{x}|z_{\leq i-2}) - I(z_{i-1}; \mathbf{x}|z_{\leq i-2}, z_i) \\
&= I(\mathbf{x}, z_i; z_{i-1}|z_{\leq i-2}) - I(\mathbf{x}; z_{i-1}|z_{\leq i-2}, z_i) \\
&= I(z_i; z_{i-1}|z_{\leq i-2}) \geq 0,
\end{aligned}
\tag{10}
\tag{11}
$$

where Eq. (10) holds by the chain rule of mutual information: $I(z_{i-1}; \mathbf{x}|z_{\leq i-2}) = I(\mathbf{x}, z_i; z_{i-1}|z_{\leq i-2}) - I(z_i; z_{i-1}|\mathbf{x}, z_{\leq i-2})$, and $I(z_i; z_{i-1}|\mathbf{x}, z_{\leq i-2}) = 0$ by the conditional independence assumption $z_i \perp z_{i-1}|\mathbf{x}, z_{\leq i-2}$. Hence if we exchange the position of $z_i$ and $z_{i-1}$ of the latent vector $z$, we can show that the new objective value after the position switch is strictly smaller

than the original one:

$$\frac{-1}{K}\left(\sum_{k=1}^{i-2} I(\mathbf{x}; z_{\leq k}) + I(\mathbf{x}; z_{\leq i-2}, z_i) + \sum_{k=i}^{K} I(\mathbf{x}; z_{\leq i})\right)$$
$$< \frac{-1}{K}\left(\sum_{k=1}^{i-2} I(\mathbf{x}; z_{\leq k}) + I(\mathbf{x}; z_{\leq i-2}, z_{i-1}) + \sum_{k=i}^{K} I(\mathbf{x}; z_{\leq i})\right) \qquad (12)$$
$$= \frac{-1}{K}\sum_{k=1}^{K} I(\mathbf{x}; z_{\leq k}).$$

Inequality (12) holds by $I(\mathbf{x}; z_i, z_{\leq i-2}) > I(\mathbf{x}; z_{i-1}, z_{\leq i-2})$. From above we can conclude that if the encoder is optimal, then the following inequalities must hold:

$$\forall i \in \{2, \cdots, K\}: \quad I(z_i; \mathbf{x}|z_{\leq i-1}) \leq I(z_{i-1}; \mathbf{x}|z_{\leq i-2}).$$

Otherwise we can exchange the dimensions to make the objective Eq. (9) smaller, which contradicts the optimally of encoder $e_\theta$. $\qquad \square$

## B   AUDIO GENERATION

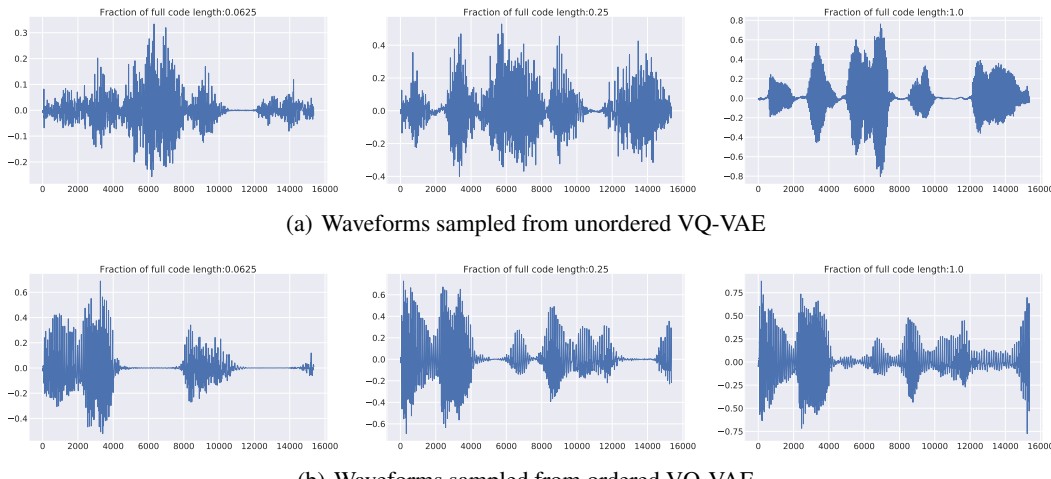

(a) Waveforms sampled from unordered VQ-VAE

(b) Waveforms sampled from ordered VQ-VAE

Figure 6: Anytime sampling with 0.0625/0.25/1.0 (**left to right**) fractions of full code length on VCTK dataset.

Our method can also be applied to audio data. We evaluate anytime autoregressive models on the VCTK dataset (Veaux et al., 2017), which consists of speech recordings from 109 different speakers. The original VQ-VAE uses an autoregressive decoder, which may cost more time than the latent autoregressive model and thus cannot be accelerated by anytime sampling. Instead, we adopt a non-autoregressive decoder inspired by the generator of WaveGan (Donahue et al., 2018). Same as images, we train a Transformer model on the latent codes.

We compare waveforms sampled from ordered vs. unordered VQ-VAEs in Fig. 6, and provide links to audio samples in Appendix G.2. By inspecting the waveforms and audio samples, we observe that the generated waveform captures the correct global structure using as few as 6.25% of the full code length, and gradually refines itself as more code dimensions are sampled. In contrast, audio samples from the unordered VQ-VAE contain considerably more noise when using truncated codes.

## C  ADDITIONAL EXPERIMENTAL RESULTS

### C.1  ORDERED VERSUS UNORDERED CODES ON VAE

To better understand the effect of order-inducing objective, we disentangle the order-inducing objective Eq. (6) with the specific implementation on VQ-VAE, such as `stop_gradient` operator and quantization. We adopt a similar order-inducing objective on standard VAE:

$$\mathbb{E}_{x \sim p_d(\mathbf{x})} \left[ \frac{1}{K} \sum_{i=1}^{K} [\mathbb{E}_{\mathbf{z} \sim q_{e_\theta}(\mathbf{z}|\mathbf{x})_{\leq i}}[- \log p_{d_\phi}(\mathbf{x}|\mathbf{z})_{\leq i}] + \mathbb{KL}(q_{e_\theta}(\mathbf{z}|\mathbf{x})_{\leq i}||p(\mathbf{z})_{\leq i})] \right], \quad (13)$$

where $q_{e_\theta}(\mathbf{z} \mid \mathbf{x})_{\leq i}$ denotes the distribution of $(z_1, z_2, \cdots, z_i, 0, \cdots, 0)^\mathsf{T} \in \mathbb{R}^K$, $\mathbf{z} \sim q_{e_\theta}(\mathbf{z} \mid \mathbf{x})$, and $p_{d_\phi}(\mathbf{x} \mid \mathbf{z})_{\leq i}$ represents the distribution of $p_{d_\phi}(\mathbf{x} \mid (z_1, z_2, \cdots, z_i, 0, \cdots, 0)^\mathsf{T} \in \mathbb{R}^K)$. We set the prior $p(\mathbf{z})_{\leq i}$ as the $i$ dimensional unit normal distribution.

We repeat the experiments on CIFAR-10 and observe similar experimental results between VQ-VAE and standard VAE. Fig. 7(a) shows that the standard VAE has irregular $\Delta(i)$, while the ordered VAE has much better ordering on latent codes. Fig. 7(b) further shows that the ordered VAE has better reconstructions under different truncated code lengths. The experimental results are consistent with Fig. 3(a) and Fig. 3(b)..

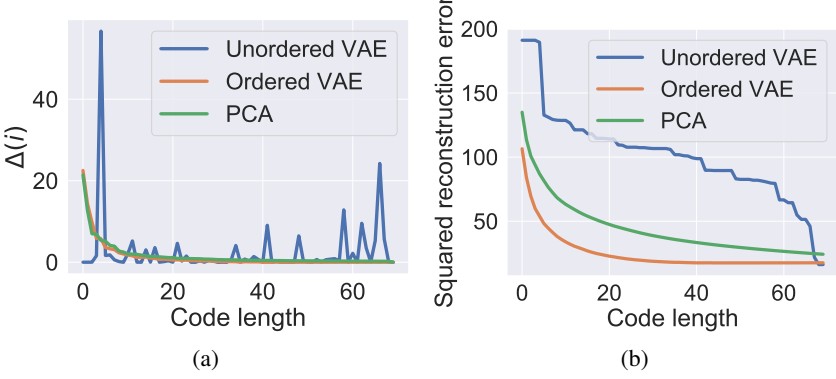

Figure 7: Ordered versus unordered code on standard VAE.

### C.2  ABLATION STUDY ON AUTOREGRESSIVE MODEL

We study the performance of different autoregressive model on ordered codes. We compare Transformer (Vaswani et al., 2017), PixelCNN (Oord et al., 2016b) and LSTM (Hochreiter & Schmidhuber, 1997). PixelCNN adopts standard 2D convolutional layers to model the conditional dependence. In order to enlarge the receptive field, PixelCNN stacks many convolutional layers. Transformer applies attention mechanism and feed forward network to model the conditional dependence of 1D sequences. LSTM uses four different types of gates to improve the long-term modeling power of the recurrent models on 1D sequences.

Transformer and LSTM can be naturally applied to the 1D channel-wise quantized codes. Since PixelCNN operates on 2D data, we reshape the 1D codes into 2D tensors. More specifically, for 70-dimensional 1D codes on CIFAR-10, we firstly pad the codes into 81 dimensions then reshape it into $9 \times 9$ tensors. Fig. 8(a) shows the FID scores of different autoregressive model on CIFAR-10 dataset. The results show that PixelCNN has inferior performance in all cases except when used with 0.2 fractions of full code length. This is because PixelCNN works well only when the input has strong local spatial correlations, but there is no spatial correlation for channel-wise quantized codes. In contrast, autoregressive models tailored for 1D sequences work better on channel-wise dequantized codes, as they have uniformly better FID scores when using 0.4/0.6/0.8/1.0 fractions of full code length.

## C.3   ABLATION STUDY ON SAMPLING DISTRIBUTION

We study the effect of different sampling distributions in order-inducing objective Eq. (6). We compare the adopted uniform distribution with the geometric distribution used in Rippel et al. (2014) on CIFAR-10, as shown in Fig. 8(b). We normalize the geometric distribution on finite indices with length $K$, i.e. $\Pr(i = k) = \frac{(1-p)^{k-1}p}{1-(1-p)^K}, i \in \{1, 2, \ldots, K\}$, and denote the normalized geometric distribution as $\text{Geo}(p)$. Note that when $p \to 0$, $\text{Geo}(p)$ recovers the uniform distribution.

Fig. 8(c) shows that OVQ-VAEs trained with all the different distributions can trade off the quality and computational budget. We find that OVQ-VAE is sensitive to the parameter of the geometric distribution. The performance of OVQ-VAE with $\text{Geo}(0.03)$ is marginally worse than the uniform distribution. But when changing the distribution to $\text{Geo}(0.1)$, the FID scores become much worse with large code length (0.6/0.8/1.0 fractions of the full code length). Since for $i \sim \text{Geo}(p)$, $\Pr(i \geq t) = (1-p)^{t-1}(1-(1-p)^{K-t+1})/(1-(1-p)^K) \leq (1-p)^{t-1}$, which indicates that the geometric distribution allocates exponentially smaller probability to code with higher index.

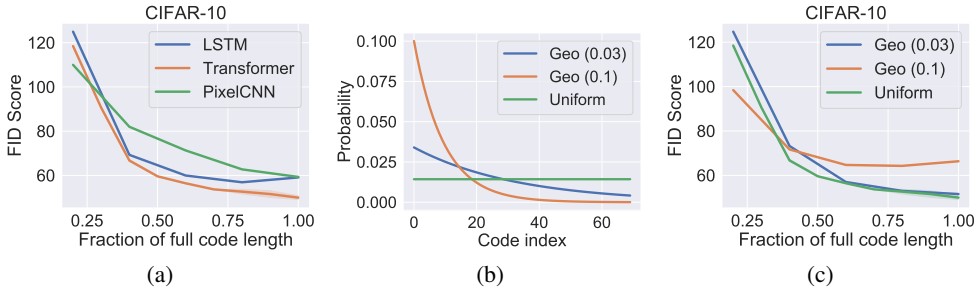

Figure 8: (a): FID scores for different autoregressive models on ordered codes; (b): Different sampling distribution for ordered codes; (c): FID scores for anytime sampling on different distributions.

## C.4   ON THE REGULARIZATION EFFECT OF ORDERED CODES

Imposing an order on latent codes improves the inductive bias for the autoregressive model to learn the codes. When using full codes on CIFAR-10 dataset, even though the OVQ-VAE has higher training error than unordered VQ-VAE, a better FID score is achieved by the ordered model. This validates the intuition that it is easier to model an image from coarse to fine. These results are corroborated by lower FID scores for the anytime model with full length codes under different number of training samples. As shown in Fig. 9(a), the ordered model has an increasingly larger FID improvement over the unordered model when the dataset becomes increasingly smaller. These results indicate that training on ordered codes has a regularization effect. We hypothesize that ordered codes capture the inductive bias of coarse-to-fine image modeling better.

## C.5   COMPARISON TO TALORED VQ-VAE

We compare the anytime sampling to the unordered VQ-VAE with tailored latent space (Tailored VQ-VAE) on CIFAR-10. The Tailored VQ-VAE has a pre-specified latent size, using the same computational budget as truncated codes of anytime sampling. For a fair comparison, we experiment with transformers on the latent space of Tailored VQ-VAE. Fig. 9(b) shows that anytime sampling always has better FID scores than Tailored VQ-VAE, except when the code is very short. We hypothesize that the learning signals from training on the full length codes with OVQ-VAE improves the quality when codes are shorter, thus demonstrating a FID improvement over the Tailored VQ-VAE. Moreover, OVQ-VAE has the additional benefit of allowing anytime sampling when the computational budget is not known in advance.

## D    TRAIN AUTOREGRESSIVE MODEL ON PCA REPRESENTATION

An alternative way to induce order on the latent space is by projecting data onto the PCA representation. However, we encounter limited success in training the autoregressive model on the top of PCA representation.

When training an autoregressive model on PCA-represented data, we observe inferior log-likelihoods. We first prepare the data by uniformly dequantizing the discrete pixel values to continuous ones. Then we project these continuous data into the PCA space by using the orthogonal projection matrix composed of singular vectors of the data covariance matrix. Note that this projection preserves the volume of the original pixel space since the projection matrix's determinant is 1, so log-likelihoods of a model trained on the raw continuous data space and the PCA projected space are comparable. We report the bits/dim (lower is better), which is computed by dividing the negative log-likelihood (log base 2) by the dimension of data. We train transformer models on the projected space and the raw data space. Surprisingly, on MNIST, the transformer model obtains 1.22 bits/dim on the projected space versus 0.80 bits/dim on the raw data, along with inferior sample quality. We hypothesize two reasons. First, models have been tuned with respect to inputs that are unlike the PCA representation, but rather on inputs such as raw pixel data. Second, PCA does not capture multiple data modalities well, unlike OVQ-VAE. Moreover, autoregressive models typically do not perform well on continuous data. In contrast, our Transformer model operates on discrete latent codes of the VQ-VAE.

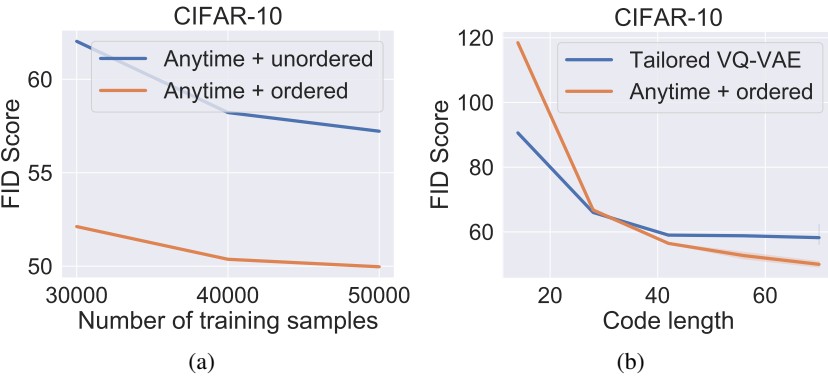

Figure 9: (a) FID scores for anytime sampling using various numbers of training samples. (b) FID scores with different computational budgets.

## E    EXTRA IMPLEMENTATION DETAILS

Our code is released via the anonymous link https://anonymous.4open.science/r/3946e9c8-8f98-4836-abc1-0f711244476d/ and included in the supplementary material as well. Below we introduce more details on network architectures and the training processes.

### E.1    NETWORKS

For MNIST dataset, the encoder has 3 convolutional layers with filter size (4,4,3) and stride (2,2,1) respectively. For CelebA and CIFAR-10 datasets, the encoder has 4 convolutional layers with filter size (4,4,4,3) and stride (2,2,2,1) respectively. The decoders for dataset above are the counterpart of the corresponding encoders. For VCTK dataset, we use a encoder that has 5 convolutional layers with a filter size of 25 and stride of 4. The activation functions are chosen to be LeakyRelu-0.2. We adopt a decoder architecture which has 4 convolutional and upsampling layers. The architecture is the same with the generator architecture in Donahue et al. (2018), except for the number of layers.

For all the datasets, we use a 6-layer Transformer decoder with an embedding size of 512, latent size of 2048, and dropout rate of 0.1. We use 8 heads in multi-head self-attention layers.

### E.2 IMAGE GENERATION

The FID scores are computed using the official code from TTUR (Heusel et al., 2017)[1] authors. We compute FID scores on CIFAR-10 and CelebA based on a total of 50000 samples and 100000 samples respectively.

We pre-train the VQ-VAE models with full code lengths for 200 epochs. Then we train the VQ-VAE models with the new objective Eq. (6) for 200 more epochs. We use the Adam optimizer with learning rate $1.0 \times 10^{-3}$ for training. We train the autoregressive model for 50 epochs on both MNIST and CIFAR-10, and 100 epochs on CelebA. We use the Adam optimizer with a learning rate of $2.0 \times 10^{-3}$ for the Transformer decoder. We select the checkpoint with the smallest validation loss on every epoch. The batch size is fixed to be 128 during all training processes.

### E.3 AUDIO GENERATION

We randomly subsample all the data points in VCTK dataset to make all audios have the same length (15360). The VQ-VAE models are pre-trained with full code length for 20 epochs, and then fine-tuned with our objective Eq. (6) for 20 more epochs. We use the Adam optimizer with learning rate $2.0 \times 10^{-4}$ for training the VQ-VAE model. We train the Transformer for 50 epochs on VCTK, use the Adam optimizer with a learning rate of $2.0 \times 10^{-3}$. We select the checkpoint with the smallest validation loss on every epoch. The batch size is fixed to be 8 for the VQ-VAE model and 128 for the Transformer during training.

## F   SAMPLES ON THE SAME PRIORITY CODE

As further illustration of the ordered encoding, we show in Fig. 10 the result of full code length sampling when the first (highest priority) discrete latent code is fixed. The fixing of the first latent code causes anytime sampling to produce a wide variety of samples that share high-level global similarities.

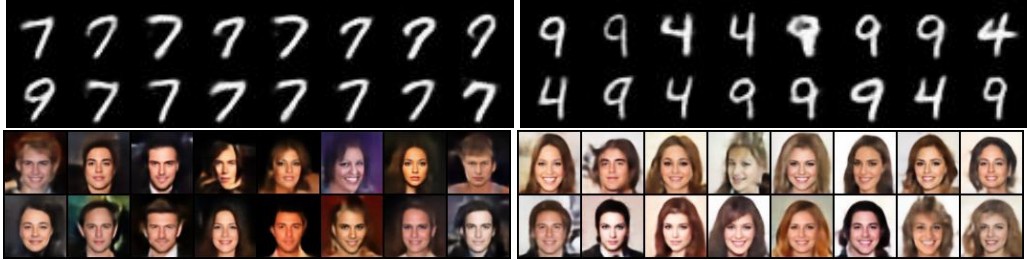

Figure 10: Samples sharing the highest priority latent code.

## G   EXTRA SAMPLES

### G.1 IMAGE SAMPLES

We show extended samples from ordered VQ-VAEs in Fig. 11, Fig. 12 and Fig. 13. For comparison, we also provide samples from unordered VQ-VAEs in Fig. 14, Fig. 15 and Fig. 16.

### G.2 AUDIO SAMPLES

We include the audio samples that are sampled from our anytime sampler in the supplementary material. The `audio` / `audio_baseline` directory contains 90 samples from ordered / unordered VQ-VAEs respectively. The fractions of full code length (0.0625, 0.25 and 1.0) used for generation are included in the names of `.wav` files.

---

[1]https://github.com/bioinf-jku/TTUR

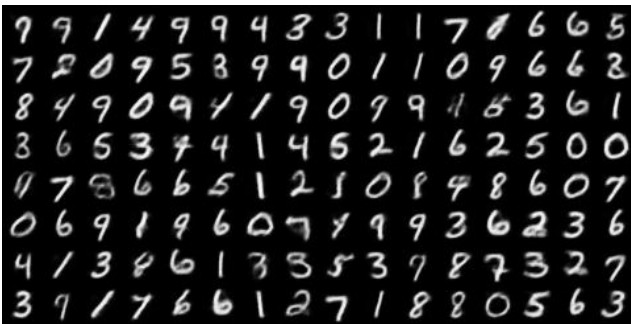

(a) 0.25 fractions of full code length

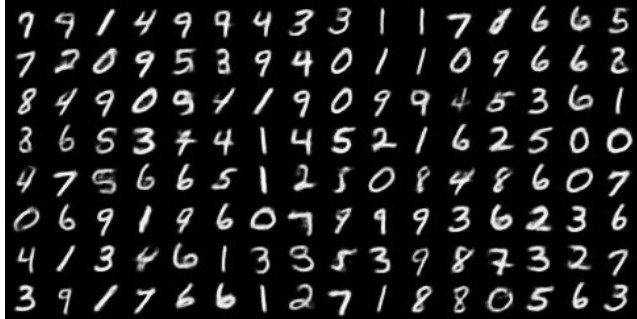

(b) 0.5 fractions of full code length

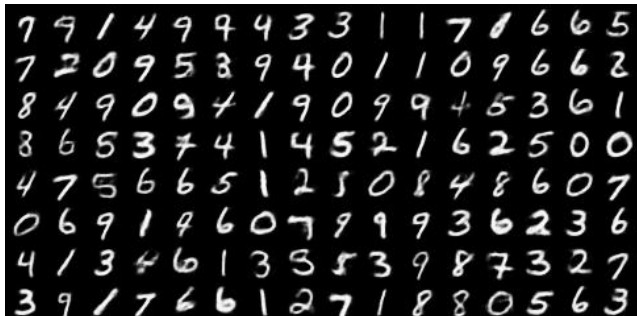

(c) 0.75 fractions of full code length

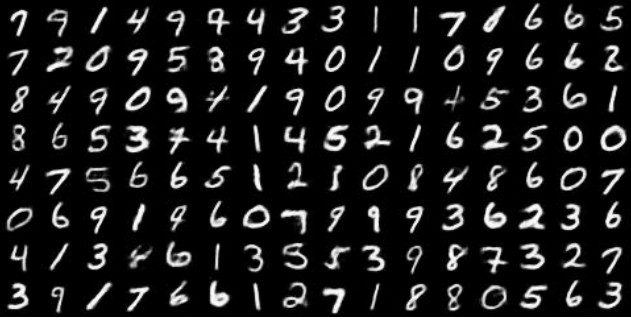

(d) 1.0 fractions of full code length

Figure 11: Anytime sampling with 0.25/0.5/0.75/1.0 (**top to bottom**) fractions of full code length from ordered VQ-VAEs on MNIST.

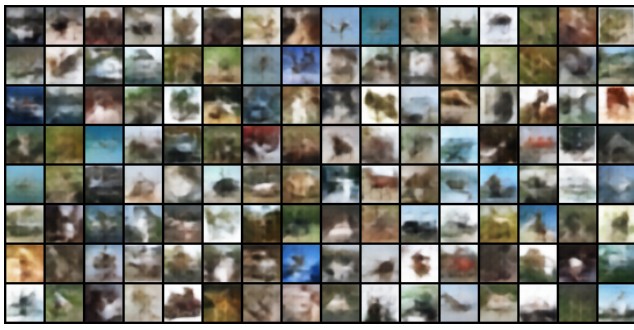

(a) 0.25 fractions of full code length

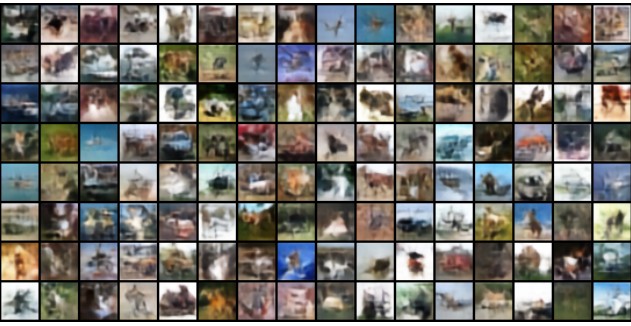

(b) 0.5 fractions of full code length

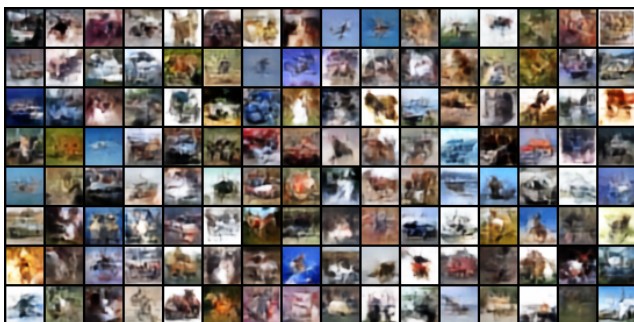

(c) 0.75 fractions of full code length

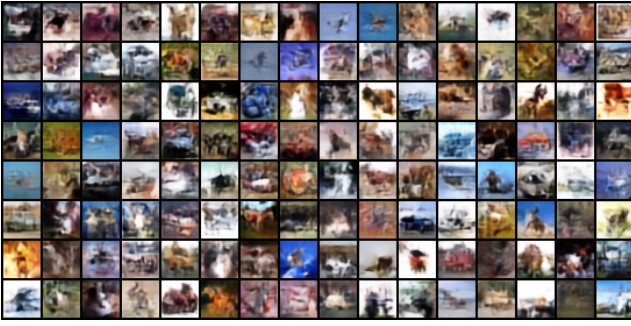

(d) 1.0 fractions of full code length

Figure 12: Anytime sampling with 0.25/0.5/0.75/1.0 (**top to bottom**) fractions of full code length from ordered VQ-VAEs on CIFAR-10.

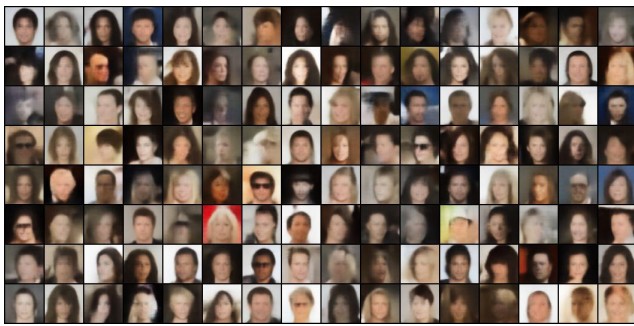

(a) 0.25 fractions of full code length

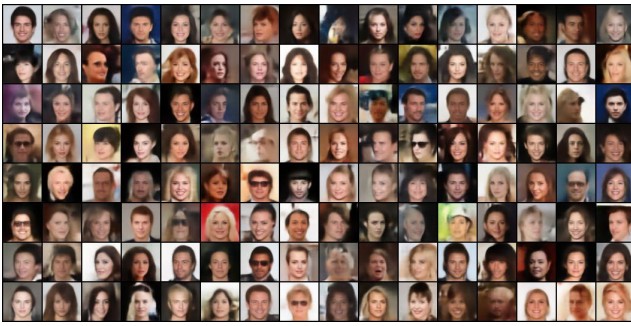

(b) 0.5 fractions of full code length

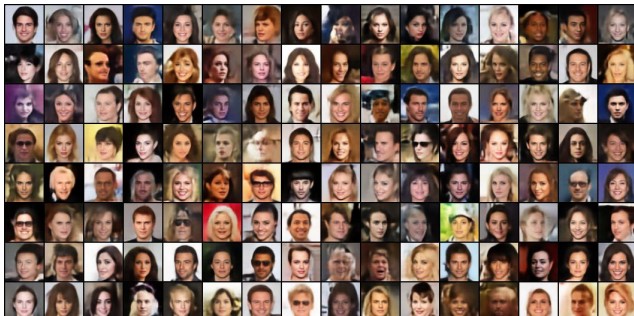

(c) 0.75 fractions of full code length

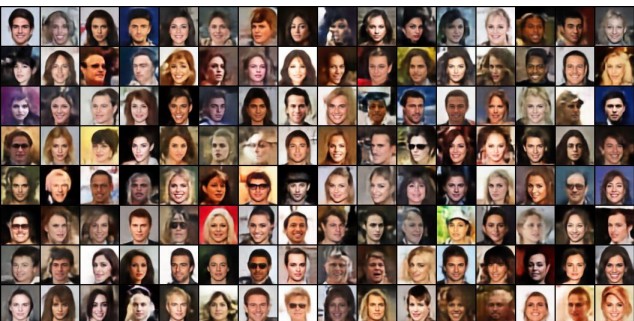

(d) 1.0 fractions of full code length

Figure 13: Anytime sampling with 0.25/0.5/0.75/1.0 (**top to bottom**) fractions of full code length from ordered VQ-VAEs on CelebA.

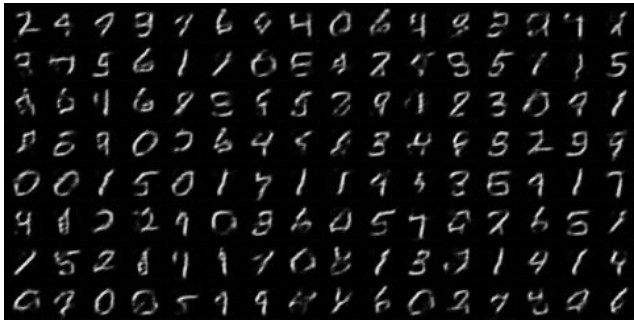

(a) 0.25 fractions of full code length

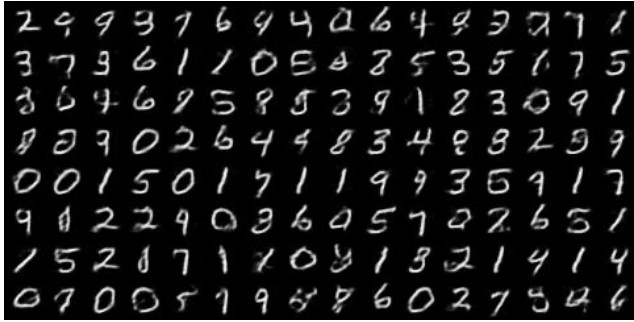

(b) 0.5 fractions of full code length

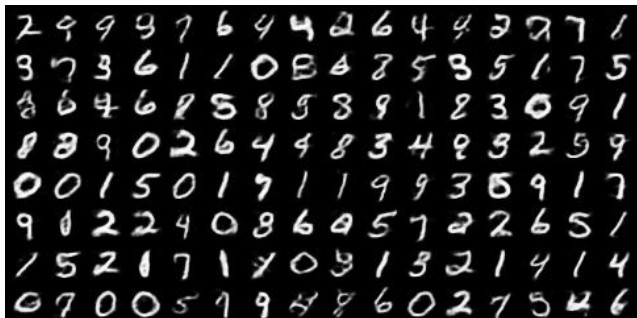

(c) 0.75 fractions of full code length

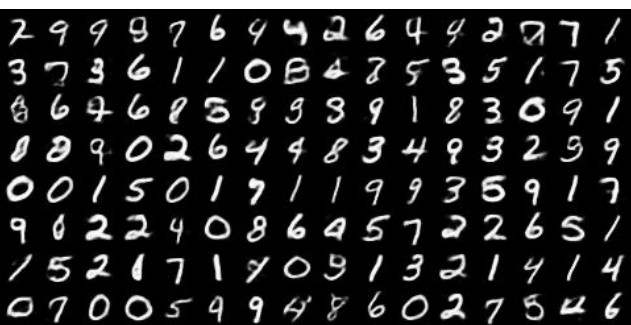

(d) 1.0 fractions of full code length

Figure 14: Anytime sampling with 0.25/0.5/0.75/1.0 (**top to bottom**) fractions of full code length from unordered VQ-VAEs on MNIST.

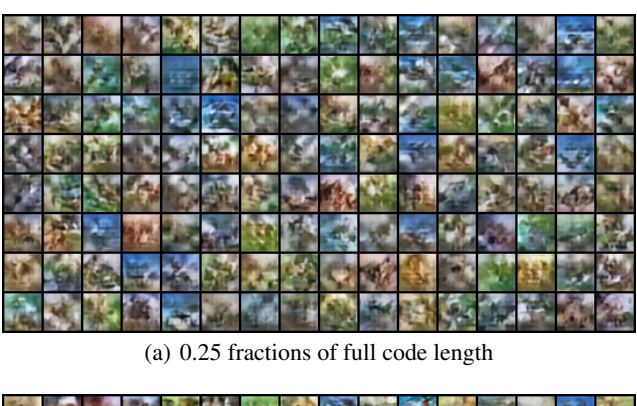

(a) 0.25 fractions of full code length

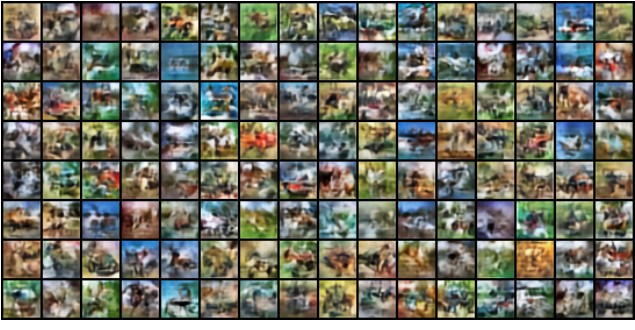

(b) 0.5 fractions of full code length

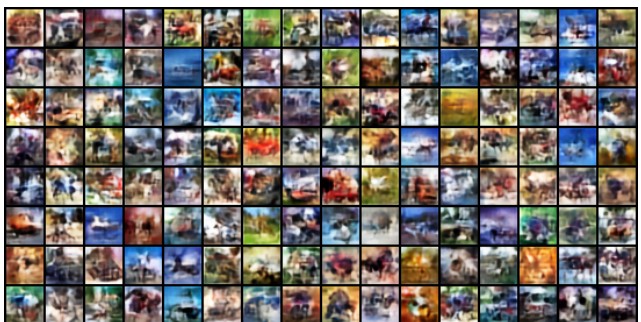

(c) 0.75 fractions of full code length

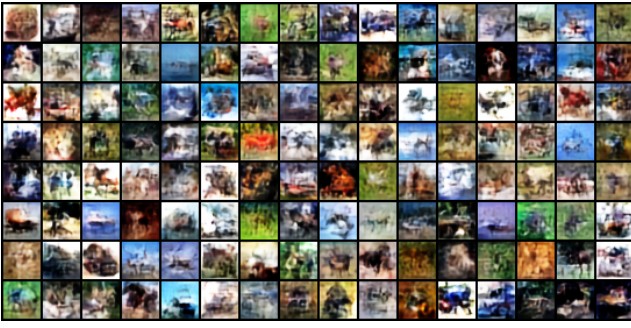

(d) 1.0 fractions of full code length

Figure 15: Anytime sampling with 0.25/0.5/0.75/1.0 (**top to bottom**) fractions of full code length from unordered VQ-VAEs on CIFAR-10.

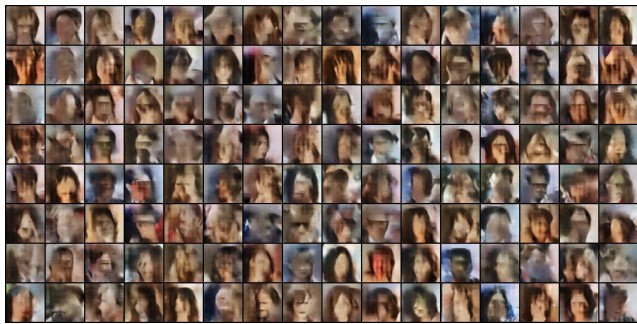

(a) 0.25 fractions of full code length

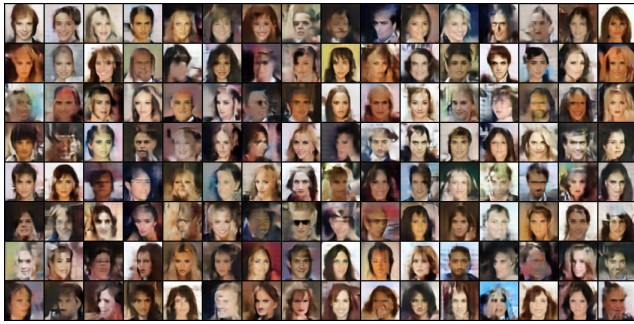

(b) 0.5 fractions of full code length

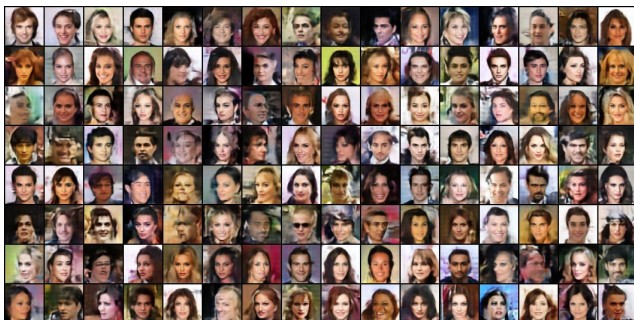

(c) 0.75 fractions of full code length

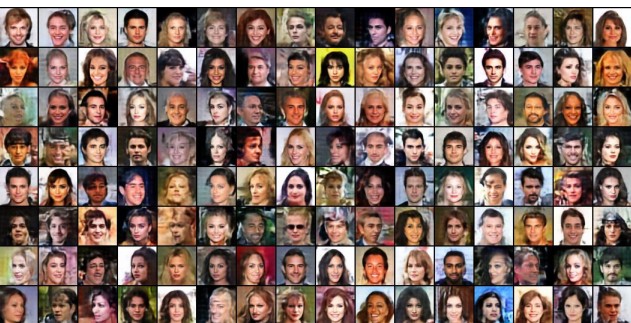

(d) 1.0 fractions of full code length

Figure 16: Anytime sampling with 0.25/0.5/0.75/1.0 (**top to bottom**) fractions of full code length from unordered VQ-VAEs on CelebA.

