# OpenReview forum: "Anytime Sampling for Autoregressive Models via Ordered Autoencoding"
_ICLR.cc/2021/Conference — ICLR 2021 Poster_

### Official Review · AnonReviewer2 · 2020-10-25
**Interesting problem & approach; experiments could be improved**

**Rating:** 7
**Confidence:** 3

**Review:**

Summary:
The authors explore the use of anytime sampling to allow sampling from autoregressive models on a fixed computational budget.
They introduce the problem formulation, and show how how to learn a latent space where the amount of information conveyed decreases consistently with a given ordering on the dimensions (in a manner reminiscent of PCA),
They then show how this can be applied to VQ-VAE to perform anytime sampling of the (discrete) latent code. The latent code can then be decoded in a fixed amount of time/computation.

Strengths:
* The authors tackle a problem that is impactful but not very well-studied in the community, and propose a novel approach to do so.
* The proposed method is well-explained.

Weaknesses:
* Experimental evalution could be more thorough.
    (1) Is there any related work on this topic (or simple baselines) that are worth comparing too?
    (2) The authors use the FID metric to evaluate the perceptual quality of their generated images, at different sampling budgets. My understanding is that FID was developed for GANs, which do not allow for a tractable likelihood calculation. However, since autogressive models do allow for it, would it make sense to use that as an evaluation metric (bits per dim is the standard metric for likelihood-based generative model)? It would be nice to see an analysis of how that metric changes as the information content in the latent code changes (which can also be measured).

Overall, I lean towards acceptance, based on the novelty of problem & approach.

---

> ### Author Response · Authors · 2020-11-25
> **Thank you for your review and suggestions**
>
> Thank you for the detailed review and thoughtful feedback. Below we address specific questions.
>
> **Q:  Is there any related work on this topic (or simple baselines) that are worth comparing too?**
>
> A: To the best of our knowledge, we are the first to focus on developing anytime sampling for autoregressive models. Prior work has tackled the slow generation of autoregressive models via improved implementations, such as the cache mechanism proposed in Ramachandran et al. (2017). Our method is orthogonal to these previous arts and can leverage them for faster sampling speed without affecting sample quality.
>
> **Q: The authors use the FID metric to evaluate the perceptual quality of their generated images, at different sampling budgets. My understanding is that FID was developed for GANs, which do not allow for a tractable likelihood calculation. However, since autoregressive models do allow for it, would it make sense to use that as an evaluation metric (bits per dim is the standard metric for likelihood-based generative model)?**
>
> A: The reviewer is correct that autoregressive models allow for tractable likelihood calculation. However, the autoregressive model in OVQ-VAE operates on the encoder's output space instead of the raw pixel space. The bits/dims are not directly comparable since the data distributions are on two different spaces.

---

### Official Review · AnonReviewer4 · 2020-10-28
**Interesting, well-motivated idea, but experiments and write-up fall a bit short**

**Rating:** 6
**Confidence:** 4

**Review:**

# Summary
The paper considers the problem of slow sampling in autoregressive generative models. Sampling in such models is sequential, so its computational cost scales with the data dimensionality. Existing work speeds up autoregressive sampling by caching activations or distilling into normalizing flows with fast sampling. Authors of this work instead propose a method that returns (approximate) samples given an arbitrary computational budget, a behaviour referred to as *anytime sampling*. The proposed model is based on VQ-VAE by van den Oord et al. (2017), where an autoregressive model is fit to a latent space of a trained discrete autoencoder, rather than to raw pixels. Authors adapt the *nested dropout* idea by Rippel et al. (2014) to encourage the discrete autoencoder to order latent dimensions by their "importance" for reconstruction. Experiments demonstrate that the ordered latent space allows to stop the autoregressive sampling process at an arbitrary latent dimension and still obtain "complete" samples. The quality of samples increases as more latent dimensions are sampled, which allows to trade sample quality for reduced computational cost.

# Strong points
- The considered problem is important, and the authors' novel framing of it is well-motivated: being able to adapt to available computational budget is potentially useful in practical applications of autoregressive generative models.
- The proposed method is well-motivated and its introduction is self-contained. The idea to use incomplete samples from an autoregressive model with a representation model that *expects* such incomplete representations is interesting and novel.
- Experiments are designed appropriately to showcase the resulting behaviour of the model. Experimental setup is explained in sufficient detail.
- Section 4.3 is well done: an easy-to-understand motivation for a proposed architectural change, with a clear empirical demonstration of its effect.

# Weak points
- Apart from channel-wise quantization, authors change the autoregressive model used in the original VQ-VAE. Authors argue that the used model is "more suitable to channel-wise quantization". While the motivation sounds reasonable, it relies on the knowledge of the inner workings of the PixelCNN and Transformer models. Including a concise description of both types of models would solidify the motivation. Moreover, an ablation experiment with alternative autoregressive models (at least PixelCNN and Transformer) would be useful to support the choice, and to learn about what autoregressive models are suitable (or not suitable) for channel-wise quantization.
- One peculiar outcome of experiments on CIFAR-10 is that a better FID score is achieved by the ordered model in comparison to the unordered one *even when sampling the full code*. This is counter-intuitive, and is not observed in CelebA experiments. Is the ordering regularizing the autoencoder somehow on the smaller CIFAR-10 dataset? Is something else going on? This phenomenon is worth exploring further to strengthen the evaluation.
- It's unclear how the proposed method compares to other methods for speeding up sampling in autoregressive models. Authors could argue that other methods do not allow for adaptive sampling and hence the comparison is not justified. But it would be useful to know how much sample quality we sacrifice with anytime sampling to achieve the speed up of the methods by Ramachandran et al. (2017) or van den Oord et al. (2018).
- Authors correctly point out that the computational cost of standard autoregressive sampling scales (at least) linearly with the input dimension. This is not the case when using *ex-post density estimation*, where it scales with the latent code dimensionality. It would be useful to know how anytime sampling compares to simply reducing the size of the latent code of the auto-encoder. In other words, what is the effect of tailoring the latent code size to a specific computational budget (if one is known in advance), as compared to anytime sampling with the same budget?
- The *ordered autoencoder* is a special case of the *nested dropout* idea proposed by Rippel et. al (2014), where $p_B$ is taken to be a uniform distribution. I wish authors explored/discussed this connection more. Why is the uniform distribution chosen, as opposed to the geometric distribution as in the original work? Why is the original motivation for using the geometric distribution not applicable in this work?
- While the theoretical analysis in 3.1.1 is interesting, I am wondering about its significance. Authors claim that Rippel et al. (2014) limit their analysis to a linear (ordered) autoencoder. However, I'd like to point out section 2.1 in the original paper, which demonstrates how exactly the training objective encourages latent dimensions to be sorted according to their "mutual information" gain. How does the offered line of reasoning relate to the one by Rippel et al., and why have authors decided to not rely on the existing analysis?
- The related work section could use some work. Authors should make it clearer that they consider a special case of the idea by Rippel et al., instead of referring to it as a "similar method", and consider moving the second paragraph to earlier sections, given how much the proposed method relies on the earlier idea. Finally, it's not clear how exactly the references in the last paragraph are relevant to this work.

# Minor points
- The remark on p7 highlights an interesting observation, but is too brief to introduce the setup properly. Consider moving it to the appendix, providing more detail on e.g. the number of principal components used or what exactly is measured in bits/dim.
- The reference for MADE in section 2 is incorrect. Change to "MADE: Masked Autoencoder for Distribution Estimation" by M. Germain, K. Gregor, I. Murray, H. Larochelle, 2015.
- The core evaluation metric (FID) should be defined beyond saying that it's "a popular metric on image quality". How exactly is it computed? And, importantly, *are we looking for a higher or a lower number?*
- The `stop_gradient` operation should be defined.
- Figure readability isn't great: a larger font (try make comparable to text font size) and thicker lines will help.
- Try to use consistent labels in figures: the proposed method is at various points referred to as a "OVQ-VAE", "Anytime + ordered", "Ours".
- References: use proper capitalization ("Gan", "pixelcnn", etc.), deal with repeated references (see Heusel 2017a/b).
- Formatting: reduced spacing above conclusion section header?
- Notation in 4.1: $z_{e,j}$ is confusing notation, consider renaming the variable to avoid having to use the comma.
- Poor phrasing: "our training is stable and more suitable for large scale training", "for the instantiation".

# Recommendation
To summarize above, I find the proposed method interesting and well-motivated. The components of the method aren't new, but the idea to combine autoregressive models with ordered representations in "ex-post density estimation" for anytime sampling is novel. Applying the nested dropout idea to VQ-VAE required some technical work. Authors have identifying a sensible set of experiments that showcase the resulting behaviour.

On the other hand, several loose threads have been left by the authors, potentially leaving the reader with questions regarding the results. More experiments (as suggested above) would help solidify the results, giving the reader a better idea about what makes the proposed method work and adding weight to the paper. The theoretical contribution could be better motivated, and put in context of existing work. The write-up itself could use some work, as also suggested above.

All in all, I find the paper in its current form to be somewhat below the standard for ICLR acceptance, unless authors address some of the points above.

# Random questions/comments
- "Progressive generation" is another potential application of this idea: a user can be presented with intermediate generations that progressively increase in quality, instead of waiting for the one final generation (think progressive JPEG).
- Have you considered sorting dimensions of a pre-trained, standard VQ-VAE? For example, by going through dimensions in turn and greedily picking the one that most reduces reconstruction error?
- In Fig 3a the OVQ-VAE curve goes up a few times. Do you explain this by optimization not fully converging?
- The changing MNIST digit class in Figure 7 is surprising. I would expect this to be stored in the highest priority part of the latent code, given its high impact on reconstruction. Is this a consistent effect?

-----

EDIT:

I thank the authors for their detailed response. I also appreciate the effort that's been put into refining the draft and undertaking the additional experiments. Most of the points I've raised have been addressed. The manuscript has been made more self-contained. Additional material on autoregressive models suitable for channel-wise quantization and the inductive bias of ordered representations is especially useful. As a result, I've increased my score.

Unfortunately, I don't feel my concerns re. paper's relationship with Rippel et. al's work have been fully addressed. It is still my understanding that the role of the analysis in 3.1 is to add rigour to the otherwise similar general argument in 2.1 of Rippel et. al. While more rigour is always good, in this case it doesn't lead to new insights, and feels tangential to the rest of the paper. As such 3.1 could be moved to the appendix, which will also help making the page limit.

In addition, I am still not fully happy with how Rippel et. al's work is discussed. A reader not fully familiar with the earlier work might get an impression that it's simply a worse version of the proposed method. It's important to make it clear that the considered method extends the work by Rippel et. al's, applying nested dropout to an autoencoder that is discrete and "variational" (arguable), and considering a different application. This would not diminish the importance of this work, but would give credit where it's due.

Overall, following author's response I am leaning towards acceptance, but will let the AC judge the importance of the points above for the final decision.

---

> ### Author Response · Authors · 2020-11-25
> **Thank you for your review and suggestions  (Response Part I)**
>
> Thank you for the detailed review and thoughtful feedback. Below we address specific questions.
>
> **Q: Authors change the autoregressive model used in the original VQ-VAE. Authors argue that the used model is "more suitable to channel-wise quantization". While the motivation sounds reasonable, it relies on the knowledge of the inner workings of the PixelCNN and Transformer models. Including a concise description of both types of models would solidify the motivation. Moreover, an ablation experiment with alternative autoregressive models (at least PixelCNN and Transformer) would be useful to support the choice, and to learn about what autoregressive models are suitable (or not suitable) for channel-wise quantization.**
>
> A: Thank you for your suggestions. We have added a concise description of PixelCNN and Transformer models in Sec 5.2. To compare different autoregressive models on channel-wise quantized codes, we experiment with Transformer, LSTM, and PixelCNN. We adopt the anytime sampling algorithm on different autoregressive models with 0.2/0.4/0.6/0.8/1.0 fractions of full code length on CIFAR10. The results show that PixelCNN has inferior performance in all cases except when used with 0.2 fractions of full code length. This is because PixelCNN works well only when the input has strong local spatial correlations, but there is no spatial correlation for channel-wise quantized codes.  In contrast, autoregressive models tailored for 1D sequences work better on channel-wise dequantized codes, as they have uniformly better FID scores when using 0.4/0.6/0.8/1.0 fractions of full code length. We have added the ablation study in Appendix C.2, and experimental results in Fig 8(a).
>
>
> **Q: One peculiar outcome of experiments on CIFAR-10 is that a better FID score is achieved by the ordered model in comparison to the unordered one even when sampling the full code. Is the ordering regularizing the autoencoder somehow on the smaller CIFAR-10 dataset?**
>
>
> A:  The reviewer is correct about the regularization of ordered codes. We validate the hypothesis by decreasing the number of training samples on CIFAR10. The FID gaps between unordered and ordered VQ-VAE using different numbers of training samples:
>
> | Number of training samples | FID gap |
> | ------ |------ |
> | 30000 | 9.91 |
> | 40000 | 7.85 |
> | 50000 | 7.25|
>
>
> We observe that the ordered model has an increasingly larger FID improvement over the unordered model when the dataset becomes increasingly smaller. These results indicate that training on ordered codes has a regularization effect. We hypothesize that ordered codes capture the inductive bias of coarse-to-fine image modeling better.
>
>
> **Q: It would be useful to know how much sample quality we sacrifice with anytime sampling to achieve the speed up of the methods by Ramachandran et al. (2017) or van den Oord et al. (2018).**
>
> A: Our method is orthogonal to both Ramachandran et al. (2017) and van den Oord et al. (2018). They can be combined together with our method without sacrificing sample quality. Specifically, Ramachandran et al. (2017) leverages caches to speed up autoregressive sampling, which can be directly applied to our autoregressive model on ordered codes without affecting sample quality. Van den Oord et al. (2018) proposes probability density distillation to distill autoregressive models into fast implicit generators. We can apply the same technique on our latent autoregressive model to allow a similar speedup. We have added this discussion to the revised draft.
>
> **Q: It would be useful to know how anytime sampling compares to simply reducing the size of the latent code of the auto-encoder. In other words, what is the effect of tailoring the latent code size to a specific computational budget (if one is known in advance), as compared to anytime sampling with the same budget?**
>
> A: As suggested by the reviewer, we compare the FID score of anytime sampling against VQ-VAE with reduced size of latent code (Tailored VQ-VAE)  on the CIFAR-10 dataset. We observe that anytime sampling always has better FID scores than tailored VAE, except when the code is very short.  We hypothesize that the learning signals from training on the full length codes with OVQ-VAE improves the quality when codes are shorter, thus demonstrating a FID improvement over the Tailored VQ-VAE. Moreover, OVQ-VAE has the additional benefit of allowing anytime sampling when the computational budget is not known in advance. We have included detailed results in Appendix C.5.

---

> > ### Author Response · Authors · 2020-11-25
> > **Response Part II**
> >
> >  **Q: Why is the uniform distribution chosen, as opposed to the geometric distribution as in the original work? Why is the original motivation for using the geometric distribution not applicable in this work?**
> >
> > A:  We include extra discussions and experiments in Appendix C.3. We show that uniform distribution is a special case of normalized geometric distribution on finite space, when the parameter in normalized geometric distribution tends to 0. Empirically, OVQ-VAEs trained with different distributions can trade off the quality and computational budget. We find that OVQ-VAE is sensitive to the parameter of the geometric distribution. The performance of OVQ-VAE with Geo(0.03) is marginally worse than the uniform distribution. But when changing the distribution to Geo(0.1), the FID scores become much worse with large code length (0.6/0.8/1.0 fractions of the full code length), since the geometric distribution allocates exponentially smaller probability to code with higher index.
> >
> >
> > **Q: While the theoretical analysis in 3.1.1 is interesting, I am wondering about its significance. Authors claim that Rippel et al. (2014) limit their analysis to a linear (ordered) autoencoder. However, I'd like to point out section 2.1 in the original paper, which demonstrates how exactly the training objective encourages latent dimensions to be sorted according to their "mutual information" gain. How does the offered line of reasoning relate to the one by Rippel et al., and why have authors decided to not rely on the existing analysis?**
> >
> > A: Thank you for pointing out the connection. The conditional mutual information in our theorem is equivalent to the mutual information gain in Rippel et al. (2014) by the chain rule of mutual information. However, we additionally prove that the mutual information gain decreases when optimizing the order-inducing objective under the general nonlinear case, which Rippel et al. (2014) did not prove.
> >
> > **Q: The related work section could use some work.**
> >
> > A: Thanks for your suggestions about the related work section. We have improved its clarity.
> >
> > ### Minor points:
> > Thank you for your correction. We have included more details for the PCA experiment in Appendix D. We have polished the writings, improved the figures, fixed the problem of references and added more descriptions for the key concepts in the revised version.
> >
> > ### Random questions/comments
> > **Q: "Progressive generation" is another potential application of this idea: a user can be presented with intermediate generations that progressively increase in quality, instead of waiting for the one final generation (think progressive JPEG).**
> >
> > A: We agree with the reviewer that progressive generation is a promising application of OVQ-VAEs and we consider this to be an important direction for future work.
> >
> > **Q: Have you considered sorting dimensions of a pre-trained, standard VQ-VAE? For example, by going through dimensions in turn and greedily picking the one that most reduces reconstruction error?**
> >
> > A: For the original VQ-VAE with spatial-wise quantization, the greedy post hoc approach will not work well, because as shown in Section 4.3 successful anytime sampling requires channel-wise quantization. Even when equipped with channel-wise quantization, the performance of this greedy post hoc method depends on how we pad the remaining dimensions when using part of the encoding, and likely will not be good because the model has never seen padded encoding during training. In contrast, our method trains the VQ-VAE with padded encodings and will generalize better to them at test time.
> >
> > **Q: In Fig 3a the OVQ-VAE curve goes up a few times. Do you explain this by optimization not fully converging?**
> >
> > A: Yes. By theorem 1, if the neural networks are fully optimized, $\Delta(i)$ would be strictly decreasing.
> >
> > **Q: The changing MNIST digit class in Figure 7 is surprising. I would expect this to be stored in the highest priority part of the latent code, given its high impact on reconstruction. Is this a consistent effect?**
> >
> > A: Yes. This observation is consistent when we fix the highest priority dimension of latent codes. The highest priority dimension may not correspond to the digit class, because we order latent code dimensions according to their reconstruction errors, and digits from the same class can have high variety and not necessarily achieve the smallest reconstruction error measured in $\ell_2$ loss. Nonetheless, samples sharing first latent codes always have similar global structures even though they may have different class labels.

---

### Official Review · AnonReviewer3 · 2020-10-30
**Straightforward paper, good enough but not super exciting or surprising**

**Rating:** 6
**Confidence:** 4

**Review:**

#### Summary

For me the paper doesn't have any really major negatives. It is straightforward and makes sense, but it also didn't have anything that jumped out to me as super exciting, surprising or interesting, hence the marginal accept.

#### Review
The paper studies the task of generative modelling with what is referred to as 'anytime sampling', that is sampling with graceful early stopping, where computation time can be traded off with sample quality. The approach taken is to adapt the VQ-VAE, imposing an ordering on latent dimensions, using a kind of dropout which effectively weights the latent dimensions differently in the objective function. The latent dimensions have a fixed ordering, and those which are early in the sequence are forced to have more explanatory power than later dimensions. Experiments are presented which demonstrate that the method basically works for images and audio, in that sample quality degrades gracefully with computational budget.

The most serious criticism I have is that I felt that little attention was given to motivating the problem. The paper seemed to assume a priori that anytime sampling is interesting and/or relevant to real-world applications. If that is so then I'm surprised there hasn't been more work addressing it. Could the authors add some more detail on their motivations for working on this, perhaps including specific examples of use-cases? Compression was mentioned very briefly near the end of the paper when discussing related work. This seems like a natural application to me, and indeed what's referred to as 'progressive decoding' in the compression community is a common feature of modern codecs. The authors don't seem to be interested in the compression use case, so why are they working on this, and why should I care?

I'm by no means an expert on the niche topic of anytime sampling, but there are a few papers which I think are sufficiently closely related that they should be mentioned. Firstly https://arxiv.org/abs/2007.06731 would fit neatly in the last paragraph of Section 6. Denoising diffusion probabilistic models (https://arxiv.org/abs/2006.11239 and https://openreview.net/forum?id=-NEXDKk8gZ) also allow trading off computational budget with sample quality, although obviously the approach is very different and this task/trade-off is not the focus of those papers.

#### Typos and other nits
 - Second paragraph of page 2, autoregressive is mis-spelled in the first line.
 - Latent vectors are transposed, why?
 - First paragraph of Section 4, last sentence says '...with small change of the original VQ-VAE...', that should either be 'with a small change to the original VQ-VAE' or 'with small changes to the original VQ-VAE'.
 - Under 'Remark' in Section 5.2.1, reference is made to 'the PCA distribution'. I don't think this is a well defined concept - PCA is not a probabilistic method and does not correspond to a probability distribution (see e.g. https://rss.onlinelibrary.wiley.com/doi/abs/10.1111/1467-9868.00196 for more detail).

---

> ### Author Response · Authors · 2020-11-25
> **Thank you for your review and suggestions**
>
> Thank you for the detailed review and thoughtful feedback. Below we address specific questions.
>
> **Q: The most serious criticism I have is that I felt that little attention was given to motivating the problem. Could the authors add some more detail on their motivations for working on this, perhaps including specific examples of use-cases?**
>
> A: We discussed the motivation of anytime sampling in the 2nd and 3rd paragraphs of introduction. Sampling from autoregressive models can be very time consuming. We hope to build an autoregressive generative model where the expensive sequential sampling process can be interrupted at any time, while still providing a valid sample. This allows us to adapt to different computation constraints without re-training the model. It is especially useful when deploying one model to multiple devices with different computational resources, that can be *unknown* at the training time. Anytime sampling method also allows the devices to dynamically adapt the model capacity to the real-time computational budget.
>
> The "Progressive generation" suggested by Reviewer 4 is another potential application of the anytime sampling model. User experiences can be significantly enhanced when presented with intermediate generations that progressively increase in quality, instead of waiting for the one final generation.
>
>
> **Q: I'm by no means an expert on the niche topic of anytime sampling, but there are a few papers which I think are sufficiently closely related that they should be mentioned. Firstly https://arxiv.org/abs/2007.06731 would fit neatly in the last paragraph of Section 6. Denoising diffusion probabilistic models (https://arxiv.org/abs/2006.11239 and https://openreview.net/forum?id=-NEXDKk8gZ) also allow trading off computational budget with sample quality, although obviously the approach is very different and this task/trade-off is not the focus of those papers.**
>
> A: Thanks for suggesting more related works. We have included them in our revised version. The primary goal of our method is to improve the adaptivity of autoregressive models, which is not studied in denoising diffusion probabilistic models. In addition, denoising generative models progressively denoise images into better quality, instead of modeling images from coarse to fine like our methods. This means that interrupting the sampling procedure of diffusion models at an early time might lead to very noisy samples, but in our case it will lead to images with corrector coarse structures and no noise which is arguably more desirable.
>
> Thank you for the corrections for the typos and nits. We have polished the writings according to your suggestions.

---

### Official Review · AnonReviewer5 · 2020-11-04
**Nice idea, but some ablation experiments are missing**

**Rating:** 6
**Confidence:** 3

**Review:**

The authors propose a novel way to trade off sample quality with computational budget in autoregressive models by performing the autoregression on the ordered latent space of a generative model. They show analytically and empirically that the tradeoff is monotonic and that it approaches the full performance at increasing computational budgets.

Major comments:
- I appreciate the application to VQ-VAEs, since in this model, the autoregressive component already plays a major role. However, I was wondering whether a toy experiment on normal VAEs (without discrete latent space) could be more easily controllable and disentangle the idea of the paper (ordered latent space autoregression) from all the problems of training discrete models (straight-through estimator, gradient stopping, etc.). Maybe something like Fig. 3 on a standard VAE could be helpful.
- If I understand correctly, the VQ-VAE in the experiments uses a latent PixelCNN, while the OVQ-VAE uses a Transformer. It is a bit hard to see how much of the performance is due to this difference. I think it would be a fairer comparison to also include a standard VQ-VAE with latent transformer, as an ablation to see how much of the effect is really due to the ordered latent space.

Minor comments:
- Sec. 1, last paragraph: autoregerssive -> autoregressive
- Sec. 6, second paragraph: Instead of uniform distribution -> Instead of the uniform distribution
- In front of Sec. 7, there seems to be a negative \vspace that shouldn't be there

Summary:
I think the idea of this paper is very intriguing and the theoretical results are convincing. However, the experiments are currently not completely satisfying, because there is no simple VAE toy experiment and no ablation study of a VQ-VAE with latent Transformer and unordered latent space.

---

> ### Author Response · Authors · 2020-11-25
> **Thank you for your review and suggestions**
>
> Thank you for the detailed review and thoughtful feedback. Below we address specific questions.
>
>
> **Q: A toy experiment on normal VAEs could be more easily controllable and disentangle the idea of the paper from all the problems of training discrete models. Maybe something like Fig. 3 on a standard VAE could be helpful.**
>
> A: Thank you for your suggestions. We repeated the experiment of ordered versus unordered codes on a standard VAE. We observe that the standard VAE has irregular $\Delta(i)$, while the ordered VAE has much better ordering on latent codes. Moreover, the ordered VAE has better reconstructions at different truncated code lengths. These experimental results are consistent with those of VQ-VAE (Fig 3). We have added the results to Appendix C.1.
>
>
> **Q: If I understand correctly, the VQ-VAE in the experiments uses a latent PixelCNN, while the OVQ-VAE uses a Transformer. It is a bit hard to see how much of the performance is due to this difference. I think it would be a fairer comparison to also include a standard VQ-VAE with latent transformer, as an ablation to see how much of the effect is really due to the ordered latent space.**
>
> A:  We have implemented VQ-VAE with a latent transformer and termed it as **Anytime+unordered** in our paper. **Anytime+unordered** models have the same architectures as **OVQ-VAE** models but are trained with the vanilla VQ-VAE loss (Eq (5)). Fig 4(a) shows that **OVQ-VAE** achieves a better FID score than **Anytime+unordered** at all code lengths, which verifies the efficacy of using the ordered latent space
>
> Thank you for the corrections in the minor comments. We have polished the writings according to your suggestions.

---

### Author Response · Authors · 2020-11-25
**A summary of updates**

We would like to thank all reviewers for high quality reviews and constructive feedback. We have revised our draft according to all the valuable comments. Major revisions are highlighted in blue in our paper. Below we summarize updates in the revised version:

### 1. More experiments

In response to R5, we have added experiments on standard VAE in Appendix C.1.  The results are consistent with VQ-VAE.
As suggested by R4, we tested on different types of autoregressive models (Appendix C.2) and various sampling distributions (Appendix C.3). In addition, we provided experimental evidence for the regularization effect of ordered codes (Appendix C.4) and the comparison to tailored auto-encoders (Appendix C.5).


### 2. Clarifying the connection & Improving the readability

We polished our writings and added more descriptions for some key concepts, such as the FID score, *stop_gradient* operation and various types of autoregressive models, as suggested by R4. In response to R4, we provided more details for the experiments on PCA-represented data in Appendix D.

In addition, we have added the missing references pointed out by R3, and included more discussion on the differences between our work and Rippel et al. (2014).

---

### Decision · Program_Chairs · 2021-01-07
**Final Decision**

**Decision:**

Accept (Poster)

**Comment:**

Many concerns raised by the reviewers have been addressed by the authors, sometimes through additional experiments. The reviewers have updated their scores in response, and all now recommend acceptance.

Like Reviewer 4, I think that the relation to nested dropout (Rippel et al. 2014) needs to be acknowledged and discussed appropriately, so I encourage the authors to carefully consider the reviewers' most recent comments about this when preparing the final version of the manuscript.

I disagree somewhat with Reviewer 3 that the motivation provided for this work is insufficient; controlling the quality/speed trade-off at inference time seems like a compelling application. So does progressive generation, as suggested by Reviewers 3 & 4. I appreciate that this is highly subjective, however. Perhaps a few more concrete examples of practical situations where such trade-offs are useful could be mentioned in the introduction.